# PRMT1 oligomerization regulates RNA-binding protein cascade to promote pancreatic cancer

Yanxia Ru[1],*, Xinyi Zhou[2],*, Xijiao Wang[2],*, Wenxin Sun[2],*, Yaohui He[2], Guosheng Hu[2], Wenjuan Li[2], Die Hu[2], Meizhi Jiang[2], Zhiming Su[1], Fengfeng Niu[1], Gang Chen[4], Jinzhang Zeng[2], Sen-Fang Sui[1,5], Wen Liu[2], Yaowang Li[1] (iD), Siming Chen[2,3] (iD)

**PRMT1 is the predominant type I protein arginine methyltransferase responsible for generating monomethylarginine and asymmetric dimethylarginine (MMA and ADMA) in various protein substrates. It regulates numerous biological processes, including RNA metabolism, mRNA splicing, DNA damage repair, and chromatin dynamics. The crystal structure of rat PRMT1 has been previously reported as a homodimer; however, higher-order oligomeric species of human PRMT1 have been observed in vivo, and their structural basis remains elusive. In this study, we present the cryo-EM structure of human PRMT1 in its oligomeric form, revealing novel interfaces crucial for PRMT1 self-assembly and normal function. PRMT1 shows a strong preference for intrinsically disordered RGG/RG motifs, which are commonly found in RNA-binding proteins, highlighting its critical role in regulating RNA metabolism. In vitro studies indicate that disrupting PRMT1 oligomerization impairs its binding affinity for RGG motif substrates, thereby reducing arginine methylation levels on these substrates. This finding emphasizes the essential role of the oligomeric state of PRMT1 in its function with RGG motif-containing RNA-binding proteins. In vivo, the loss of PRMT1 oligomerization leads to decreased global ADMA levels in pancreatic ductal adenocarcinoma (PDAC) cells and inhibits PDAC tumor growth. Collectively, we propose that PRMT1 oligomerization, rather than mere dimerization, is sufficient for PRMT1-driven PDAC tumor growth, offering novel insights into the potential therapeutic targeting of PRMT1 oligomeric forms in PDAC.**

## Introduction

Protein arginine methylation is a ubiquitous post-translational modification in eukaryotic cells, catalyzed by protein arginine methyltransferases (PRMTs) (1). Currently, nine PRMT members have been identified in humans, which are classified into three distinct types based on their final methyl-arginine products. Type I PRMTs, including PRMT1, PRMT3, PRMT4, PRMT6, and PRMT8, facilitate the formation of asymmetric dimethylarginine (ADMA). In contrast, type II PRMTs (PRMT5 and PRMT9) catalyze the production of symmetric dimethylarginine (SDMA), although type III PRMT (PRMT7) is exclusively responsible for the generation of monomethylarginine (MMA), which serves as an intermediate in both types I and II PRMT reactions (1, 2). PRMTs transfer a methyl group from S-adenosylmethionine (SAM) to their histone or non-histone substrates, thereby modifying their stability, localization, or activity (3). These modifications profoundly influence various cellular processes, including RNA metabolism, DNA repair, chromatin dynamics, and liquid-liquid phase separation (LLPS) (1, 3, 4, 5, 6, 7). Therefore, it is not surprising that the expression and activity of PRMTs are tightly regulated, and aberrant arginine methylation is frequently linked to various health disorders, including cardiovascular, neurodegenerative, and cancer-related diseases (6, 8).

PRMT1 is the major type I protein arginine methyltransferase in mammalian cells, accounting for 85% of cellular PRMT activity (9). Knockout of PRMT1 in mice results in embryonic lethality, underscoring its critical role in embryogenesis and development (10). Furthermore, aberrant expression of PRMT1 has been documented in several cancer types, including breast, colorectal, and pancreatic cancers (11, 12, 13, 14, 15, 16). Inhibiting PRMT1 activity effectively restricts tumor cell growth, positioning it as an attractive target for therapeutic intervention (6, 8). The crystal structure of rat PRMT1 reveals that it adopts a head-to-tail homodimeric architecture, which is facilitated through its dimerization arm (17). Mutations or removal of the dimer arm disrupt PRMT1 dimerization, leading to a loss of its activity (17). These findings underscore the critical importance of the dimeric state for PRMT1's proper biological functions. Actually, dimerization is a feature conserved across all type I PRMTs and is essential for their normal catalytic activity (18).

Although it is well recognized that dimerization is required for PRMT1 to perform its biological functions, a series of studies

[1]School of Life Sciences, Southern University of Science and Technology, Shenzhen, China   [2]School of Pharmaceutical Sciences, Fujian Provincial Key Laboratory of Innovative Drug Target Research, Xiamen University, Xiamen, China   [3]Shenzhen Research Institute of Xiamen University, Shenzhen, China   [4]The First Affiliated Hospital of Wenzhou Medical University, Wenzhou, China   [5]Beijing Frontier Research Center for Biological Structures, Beijing Advanced Innovation Center for Structural Biology, School of Life Sciences, Tsinghua University, Beijing, China

Correspondence: simingchen@xmu.edu.cn; liyw@sustech.edu.cn; w2liu@xmu.edu.cn; suisf@mail.tsinghua.edu.cn
*Yanxia Ru, Xinyi Zhou, Xijiao Wang, and Wenxin Sun contributed equally to this work

indicate that PRMT1 forms large oligomeric species in vivo ([17], [19], [20], [21], [22], [23], [24], [25]). For instance, either in undifferentiated human primary keratinocytes or murine erythroleukemia (MEL) cells, both endogenous and overexpressed PRMT1 exhibit similar elution behavior to elute in fractions with a molecular mass ranging from 250 to 600 kD ([19], [20]). In addition, although the crystal structure of rat PRMT1 has previously been reported as a homodimer, higher-order oligomeric forms of rat PRMT1 are also observed in solution ([17]). Collectively, these findings strongly imply that PRMT1 may function in a higher-order oligomeric state in vivo. However, whether these higher oligomeric forms confer distinct biological effects compared with the dimeric form remains unclear.

Motifs rich in arginines and glycines were discovered decades ago and were termed the RGG/RG motifs, which are commonly found in RNA-binding proteins (RBPs) ([26]). More than 1,000 human proteins contain the RGG/RG motifs, and these proteins regulate numerous biological processes such as RNA metabolism, mRNA splicing, DNA damage repair, mRNA translation, signal transduction, and LLPS ([26], [27], [28], [29]). It is now well accepted that arginines within RGG/RG motifs are preferred sites for methylation by PRMTs, in which arginine methylation affects direct interactions of RGG motif proteins with RNAs or proteins, their participation in LLPS, and protein localization ([7], [26], [30], [31]). Nonetheless, how PRMT1 recognizes its specific substrates and to what degree this recognition is regulated by its oligomeric state remain elusive. Given that multiple RGG/RG repeats are frequently observed in RBPs, such as the C-terminus of nucleolin (RGGGFGGRGGFGDRGGRGGGRGG) harbor five RGG/RG repeats (underlined), and considering that PRMT1 may function in larger oligomers than dimers in vivo, we hypothesize that PRMT1 uses its oligomeric state to recognize RGG motif substrates, particularly those with multiple RGG/RG repeats, through multivalent interactions, thereby enhancing binding affinity and selectivity.

Here, we present the cryo-electron microscopy (cryo-EM) structure of human PRMT1, which adopts a homo-oligomeric architecture featuring a novel helical polymer. Biochemical assays and mutagenesis analyses, combined with size exclusion chromatography (SEC), reveal new interfaces that are critical for PRMT1 self-assembly and normal function. Disrupting PRMT1 oligomerization affects its binding affinity for substrates containing RGG motifs and reduces the levels of ADMA modifications on these substrates. Furthermore, we demonstrate that the oligomeric state is essential for PRMT1's role in facilitating the proliferation of pancreatic ductal adenocarcinoma (PDAC) cells. The loss of PRMT1 oligomerization results in decreased global ADMA levels and suppresses PDAC tumor growth. Mechanistically, our transcriptomic and proteomic data indicate that disrupting PRMT1 oligomerization impairs RNA metabolism by reducing its binding affinity for several RBPs containing RGG motifs. This finding is further supported by previous studies that highlighted the PRMT1-dependent regulation of RNA metabolism in sustaining PDAC ([14], [32]). Overall, our findings provide structural insights into the oligomeric scaffold of PRMT1, facilitating its recognition of substrates, particularly those RBPs containing multiple RGG repeats, through multivalent interactions, and offer novel perspectives on the potential therapeutic targeting of PRMT1 oligomeric forms in PDAC.

# Results

## Cryo-EM structure of the human PRMT1 helical polymer

To determine whether PRMT1 can form higher-order oligomers in solution, we subjected purified recombinant human PRMT1 protein to gel filtration chromatography. The PRMT1 protein eluted as a broad peak with a molecular weight ranging from 200 to 440 kD on a Superose 6 Increase 10/300GL column, indicating its self-association into oligomeric species under the gel filtration buffer conditions (Fig S1A). This observation is consistent with previous findings regarding human PRMT1 ([17]). To further elucidate the properties of PRMT1 self-association in vitro, we assessed the concentration-dependent behavior of PRMT1 by negative stain experiments (Fig 1A). The negative-stain images revealed that PRMT1 can self-assemble into helical polymers in a concentration-dependent manner. The tendency of PRMT1 to form higher-order oligomers increased when the protein concentration exceeded 0.1 mg/ml. Short helical polymers emerged when the concentration was up to 0.2 mg/ml, and the longer helical polymer form dominated after the concentration was over 0.4 mg/ml. Overall, the negative-stain images indicated that the formation of PRMT1 helical polymers is highly dependent on concentration.

Next, we aimed to determine the experimental structure of PRMT1 oligomers using cryo-EM to address existing knowledge gaps between the structural and functional aspects of human PRMT1. Cryo-EM images collected using a Titan Krios microscope (see Table 1 for collection parameters) revealed linear oligomers of varying lengths (Fig 1B). One representative micrograph indicates that, although some polymers lacked orderly assembly and exhibited heterogeneity, the PRMT1 helical polymers were generally well-distributed (Fig 1B). Therefore, homogeneous and highly ordered helical polymers were further isolated through classification. In addition, helical symmetry, a key parameter in helical reconstruction, was estimated using a program developed by the author, which is available for download from GitHub (https://github.com/lyw8120/picker). The cryo-EM map of PRMT1 was reconstructed to a resolution of 3.68 Å (Fig S2). The local resolution is sufficient to determine that the repeating unit of the PRMT1 oligomer is a dimer of PRMT1 (colored in green and magenta), with an approximate diameter of 114 Å (Fig 1C). Thus, the thematically helical polymer was generated following the calculated helical symmetry from cryo-EM data (Fig 1D).

## Multiple types of contacts stabilize the PRMT1 dimer

As illustrated in Fig 2A, the identified PRMT1 dimer model serves as a repeating unit within the PRMT1 oligomer. The two identical interfaces in the dimer can be directly visualized from the interior perspective of the helical polymer. These interfaces are consistent for each pair of PRMT1 monomers (blue to magenta and magenta to blue in Fig 2A). PRMT1 adopts a head-to-tail homodimeric orientation, where the head of one monomer contacts the tail of another. Specifically, a loop from one monomer is inserted into the hydrophobic pocket of the other, with amino acid residues W215, W216, V219, Y220, F222, D223, and M224 from the loop interacting with the hydrophobic pocket of the adjacent monomer.

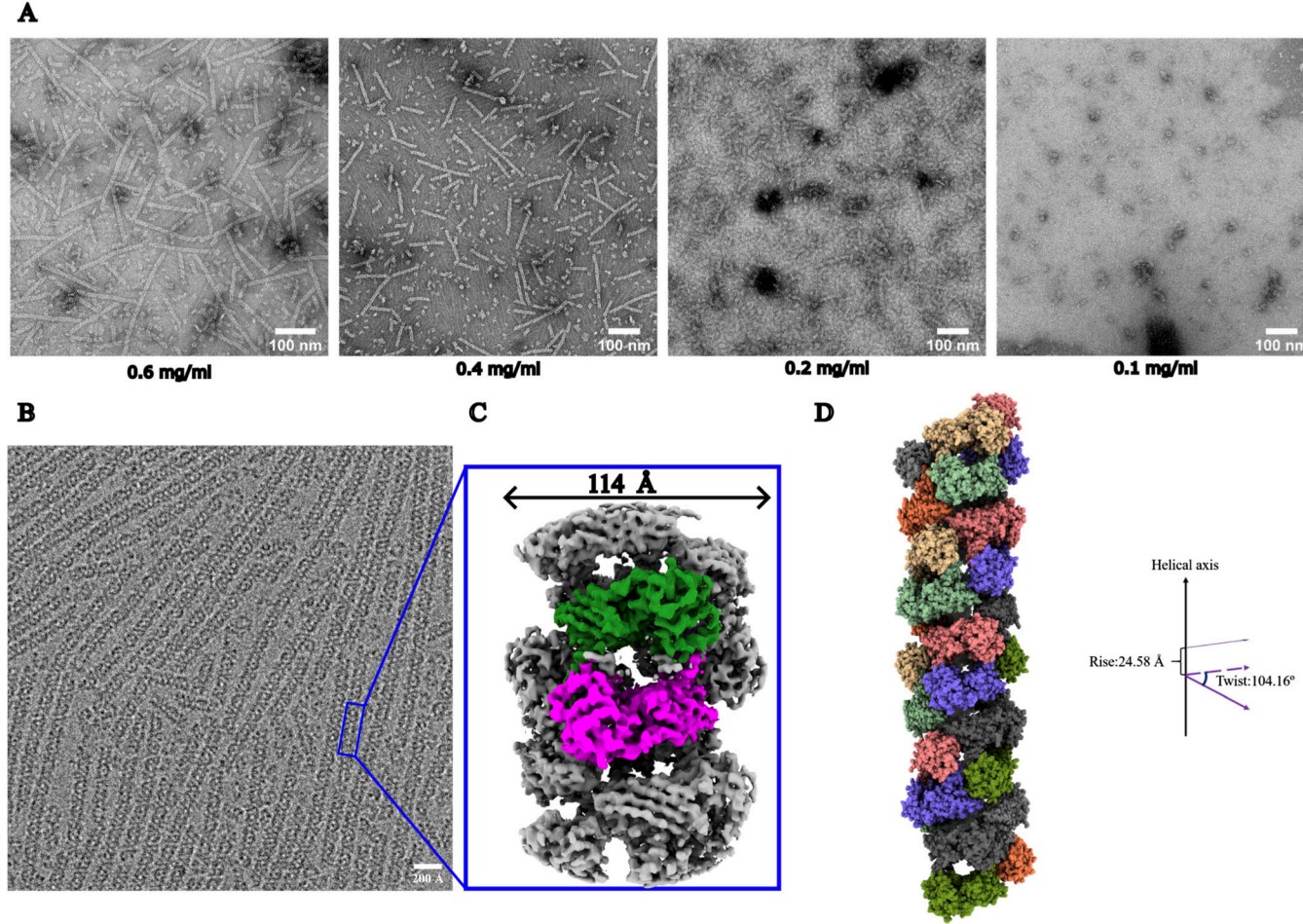

**Figure 1. Concentration-dependent PRMT1 helical polymer formation and its helical architecture.**
**(A)** The negative stain data of four different concentrations of PRMT1, ranging from 0.6 to 0.1 mg/ml. **(B)** A cryo-electron microscopy (cryo-EM) micrograph of 0.6 mg/ml PRMT1 was shown, in which almost all PRMT1 monomers assembled into helical polymers. **(C)** The helical polymer was solved at 3.68 Å, and the PRMT1 colored green and magenta are monomers; the helical polymer diameter is about 114 Å, and its basic unit is a PRMT1 dimer. **(D)** PRMT1 dimers are assembled into helical polymer one by one at the position after rotating around the helical axis by 104.16o and moving along the direction of the helical axis 24.58 Å distance simultaneously.

Although the interface between the two monomers is primarily composed of several hydrophobic contacts, electrostatic interactions also contribute to the stabilization of the PRMT1 dimer. Fig 2B illustrates various interactions between amino acid residues at the interface of the two monomers. A cation-π interaction between K68 and W215 associated with a hydrogen bond between L67 (or T73) and Y220, exerts strong interaction on the left side of the hydrophobic pocket. A typical hydrophobic contact that F222 positions between M104 and F105 and another hydrogen bond forms between N133 and D223 demonstrated on the right side of the hydrophobic pocket. In summary, the cation-π and hydrogen bond interactions on the left, the hydrophobic contacts in the center, and the additional hydrogen bond on the right collectively stabilize the PRMT1 dimer interface, with W215, Y220, and F222 serving as key residues in this stabilization.

### A unique pattern of contacts is critical to assemble PRMT1 helical polymer

The helical polymer formation depends on the connections between dimers, which serve as the repeating units of the PRMT1

oligomer. The positions of the three dimers (A, B, and C) are illustrated in Fig 3A. Two identical interfaces (interfaces 1 and 2) exist between dimers A and B, alongside a weaker contact between dimers A and C (interface 3). In interface 3, T327 interacts with T327. Y280 from dimer A (shown in green) is inserted into the electrostatic pocket of dimer B (shown in blue), and H296 from dimer B is inserted into the electrostatic pocket of dimer A in the interface 1/interface 2 (Fig 3A). These residues are depicted in greater detail in Fig 3B, where Y322 forms hydrogen bonds with F81, Y280 forms hydrogen bonds with H82 (or N83), and N278 interacts with H296. Contacts at these interfaces strongly couple the dimers, promoting their assembly into helical polymers. Notably, the hydrogen bond pairs, N278 with H296 and Y280 with H82 (or N83), along with adjacent residues, create two "U-shaped" pockets that fit into each other in a face-to-face arrangement (Fig 3C).

The current structure allowed us to design various PRMT1 mutations that selectively disrupt PRMT1 oligomers into dimers or monomers in solution. Different types of interfaces are crucial for forming PRMT1 helical polymers. The first interface occurs between PRMT1 monomers, where the residues W215, Y220, and F222 play key

**Life Science Alliance**

**Table 1. Cryo-EM data collection, refinement, and validation statistics.**

| | PRMT1 (EMD-61691) (PDB 9JP0) |
|---|---|
| Data collection and processing | |
| Magnification | 130 K |
| Voltage (kV) | 300 |
| Flux (e-/pix/sce) | 20 |
| Frames per exposure | 32 |
| Electron exposure (e-/Å$^2$) | 50 |
| Pixel size (Å) | 0.92 |
| Defocus range (μm) | 1.0–2.5 |
| Micrographs collected | 1,675 |
| Final particle images (no.) | 150,914 |
| Resolution at 0.143 FSC (Å) | 3.68 Å |
| Refinement | |
| Initial model used | AlphaFold2 prediction |
| Model resolution (Å) | 3.5 |
| FSC threshold | 0.143 |
| Map sharpening B factor (Å$^2$) | −150 |
| Model composition | |
| Non-hydrogen atoms | 25,322 |
| Protein residues | 3,120 |
| Ligands | 0 |
| B factors (Å$^2$) | |
| Protein | 154.11 |
| Ligand | 0 |
| R.m.s. deviations | |
| Bond lengths (Å) | 0.004 |
| Bond angles (°) | 0.645 |
| Validation | |
| MolProbity score | 1.90 |
| Clashscore | 9.34 |
| Poor rotamers (%) | 0 |
| Ramachandran plot | |
| Favored (%) | 94.00 |
| Allowed (%) | 6.00 |
| Disallowed (%) | 0 |

roles in stabilizing the PRMT1 dimer (Fig 2B). The second interface is formed between dimers, in which the residues Y280 and H296, located within the "U-shaped" pockets, are vital for assembling the PRMT1 dimer into helical polymers (Fig 3C). To confirm the contribution of these amino acid residues to the stabilization of the dimeric or oligomeric state of PRMT1 in solution, we introduced mutations into these residues to generate distinct oligomeric forms of PRMT1. For instance, the mutants PRMT1$^{W215A/Y220A/F222A}$ and PRMT1$^{Y280A/H296A/T327A}$ were generated to produce the monomeric and dimeric states of PRMT1, respectively. To verify the accuracy of

the human PRMT1 structure, gel filtration experiments were performed to examine the effects of these amino acid mutations on the oligomerization of the PRMT1. Consistent with our cryo-EM structure of human PRMT1, we observed that purified WT PRMT1 eluted as an oligomer on a SEC column, as previously reported ([17]). Conversely, under the same experimental conditions, PRMT1$^{W215A/Y220A/F222A}$ and PRMT1$^{Y280A/H296A/T327A}$ behaved as a monomer and a dimer, respectively (Fig S3A). Notably, the single mutations Y280A or H296A in PRMT1 sufficiently disrupted the PRMT1 oligomer into a dimer (Fig S3B).

Consistent with the gel filtration results, negative stain images revealed that the PRMT1$^{W215A/Y220A/F222A}$ mutant was unable to assemble into helical polymers and could not form dimers (Fig S3C). In addition, 2D classification of these negative stain images further demonstrated that the PRMT1$^{W215A/Y220A/F222A}$ mutant exists as a monomer (Fig S3C). However, the negative stain images and additional 2D classification indicated that the PRMT1$^{Y280A/H296A/T327A}$ mutant is incapable of forming helical polymers but retains the dimeric form (Fig S3C).

Finally, a two-tag co-immunoprecipitation analysis was conducted to verify the critical role of the aforementioned amino acid residues in the formation of PRMT1 oligomers (Fig S3D). In this assay, equal amounts of WT or mutant FLAG-PRMT1 and MYC-PRMT1 plasmids were co-transfected into HEK293T cells (Fig S3D). Subsequently, nuclear and cytoplasmic extracts were prepared and separately bound to anti-FLAG resin. MYC-PRMT1 binding, via protein oligomerization or dimerization, was assessed using anti-MYC antibodies. Consistently, mutations of W215, Y220, and F222 completely abolished PRMT1 oligomerization and dimerization; however, the Y280, H296, and T327 mutations allowed PRMT1 dimerization to persist; whereas, oligomerization was lost (Fig S3E and F). Thus, W215-, Y220-, and F222-mediated dimer stabilization, along with Y280-, H296-, and T327-mediated oligomer stabilization, were validated in the context of full-length PRMT1 protein in HEK293T cells (Fig S3D–F). In subsequent studies, unless otherwise specified, we refer to WT PRMT1$^{WT}$ as the oligomeric form of PRMT1 (hereafter referred to as PRMT1$^{Oligomer}$), the PRMT1$^{Y280A/H296A/T327A}$ mutant as the dimeric form of PRMT1 (hereafter referred to as PRMT1$^{Dimer}$), and the PRMT1$^{W215A/Y220A/F222A}$ mutant as the monomeric state of PRMT1 (hereafter referred to as PRMT1$^{Monomer}$) to investigate the effects of different PRMT1 oligomeric states on its biological functions.

### Oligomerization is required for stable binding of PRMT1 to substrates containing RGG motifs

The arginine- and glycine-rich (RGG/RG) motifs are preferentially methylated by PRMT1, with many previously identified PRMT1 substrates containing multiple tandem RGG motifs ([5], [6], [26], [30], [31]). Given that oligomers larger than dimers have been observed for PRMT1 in vivo, an efficient mechanism to enhance substrate binding is to increase binding avidity through protein oligomerization ([24], [34]). In this regard, the binding of PRMT1 to substrates containing multiple tandem RGG motifs may be facilitated by the oligomeric structural platform provided by PRMT1. To validate this hypothesis, we initially examined the impact of distinct oligomeric states on PRMT1 binding to a range of its substrates, including hnRNPA1,

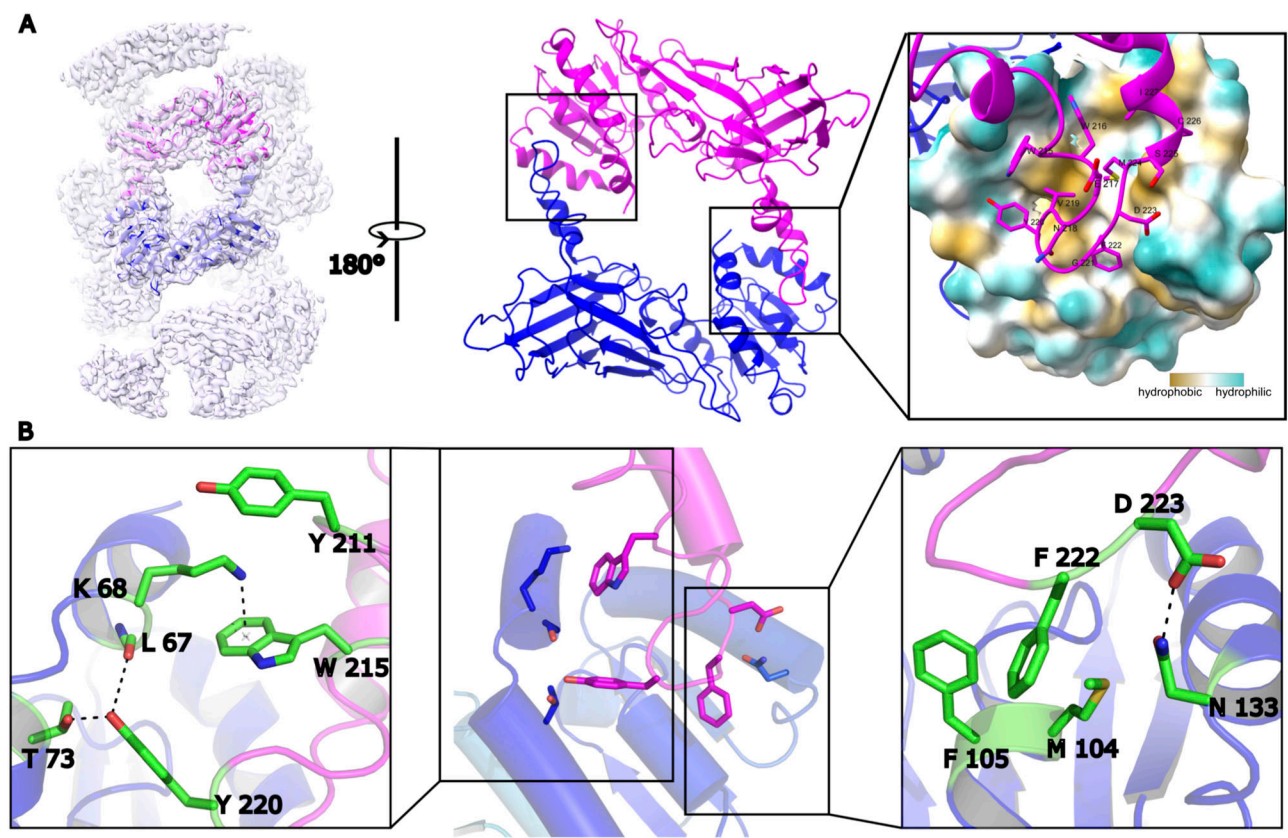

**Figure 2. The formation of the PRMT1 dimer.**
**(A)** The head-tail connection formed a dimer, making it easy to find that a loop in the tail region of one monomer was inserted into the head region of another monomer from the view of the helical polymer interior. The head region of PRMT1 in the interface is a hydrophobic pocket to hold the tail region of another PRMT1. **(B)** the contacts in the interface were shown as two parts: a cation-pi contact between W215 and K68 was found, and a hydrogen bond can be formed between Y220 and L67 (or T73) in the left part; the hydrophobic contact was formed that F222 inserts into the center of M104 and F105 and a hydrogen bond was formed between D223 and N133 in the correct part.

hnRNPA2, hnRNPK, METTL14, GAR1, FUS, Nucleolin, and Fibrillarin, all of which possess more than three tandem RGG motifs and are known to be methylated by PRMT1 in multiple cell lines (Fig 4A) (35, 36, 37, 38, 39, 40, 41, 42, 43, 44). To achieve this objective, we conducted in vitro pull-down assays to compare the binding affinity of PRMT1 for the aforementioned substrates in the context of PRMT1$^{oligomers}$, PRMT1$^{dimers}$, and PRMT1$^{monomers}$. The GST-tagged and FLAG-tagged RGG motif substrates were expressed and purified from *E. coli* for these assays. Coomassie blue staining clearly demonstrated that the oligomeric form of PRMT1 exhibited a markedly higher binding affinity for these RGG substrates compared with its dimeric and monomeric forms (Fig 4B–H). Domain mapping using a pull-down assay further demonstrated that the C-terminal RGG-rich region of hnRNPA1 is responsible for the stable binding of PRMT1 (Fig S4A and B). These findings collectively support the notion that the avidity effect mediated by oligomers influences PRMT1 binding to its RGG region-containing substrates.

We then assessed the extent to which the oligomeric state influences the binding of PRMT1 to substrates containing the RGG region in living cells. FLAG-labeled RGG motif substrates, including hnRNPA1, FUS, hnRNPK, Fibrillarin, GAR1, TAF15, Nucleolin, and RBFOX2, were co-transfected with distinct oligomeric states of MYC-tagged PRMT1, namely MYC-PRMT1$^{oligomer}$, MYC-PRMT1$^{dimer}$, and

MYC-PRMT1$^{monomer}$, into HEK293T cells. RGG motif-containing substrates were captured using anti-FLAG resin, and the differential binding of the various oligomeric states of PRMT1 to these substrates was eluted with FLAG peptide and visualized by Western blotting. Consistent with the previously described in vitro pull-down results (Fig 4B–H), the binding of RGG motif substrates was diminished for PRMT1$^{monomer}$ and PRMT1$^{dimer}$ compared with PRMT1$^{oligomer}$ in HEK293T cells, highlighting emphasizing the crucial role of the avidity effect mediated by PRMT1 oligomerization in RGG motif substrates binding in living cells (Fig 5A–H).

The mechanism by which PRMT1 specifically methylates certain arginine residues within RGG regions remains unresolved. To address this, we aimed to determine the cryo-EM structure of PRMT1 in complex with RGG-rich substrates to elucidate the structural basis for its recognition of RGG motifs. Fibrillarin and hnRNPA1 were selected as model RGG-rich substrates, and gel filtration experiments were conducted to assess whether PRMT1 could co-elute with either fibrillarin or hnRNPA1 on a gel filtration column. The results indicated that PRMT1 forms stable complexes with both hnRNPA1 and fibrillarin on a SEC column (Figs 4I and J and S4C and D). Finally, single-particle cryo-EM analysis of the PRMT1-hnRNPA1 and PRMT1-fibrillarin complexes confirmed that PRMT1 preferentially adopts a hexameric form in both complexes (Figs 4K and L and S4E and F), further

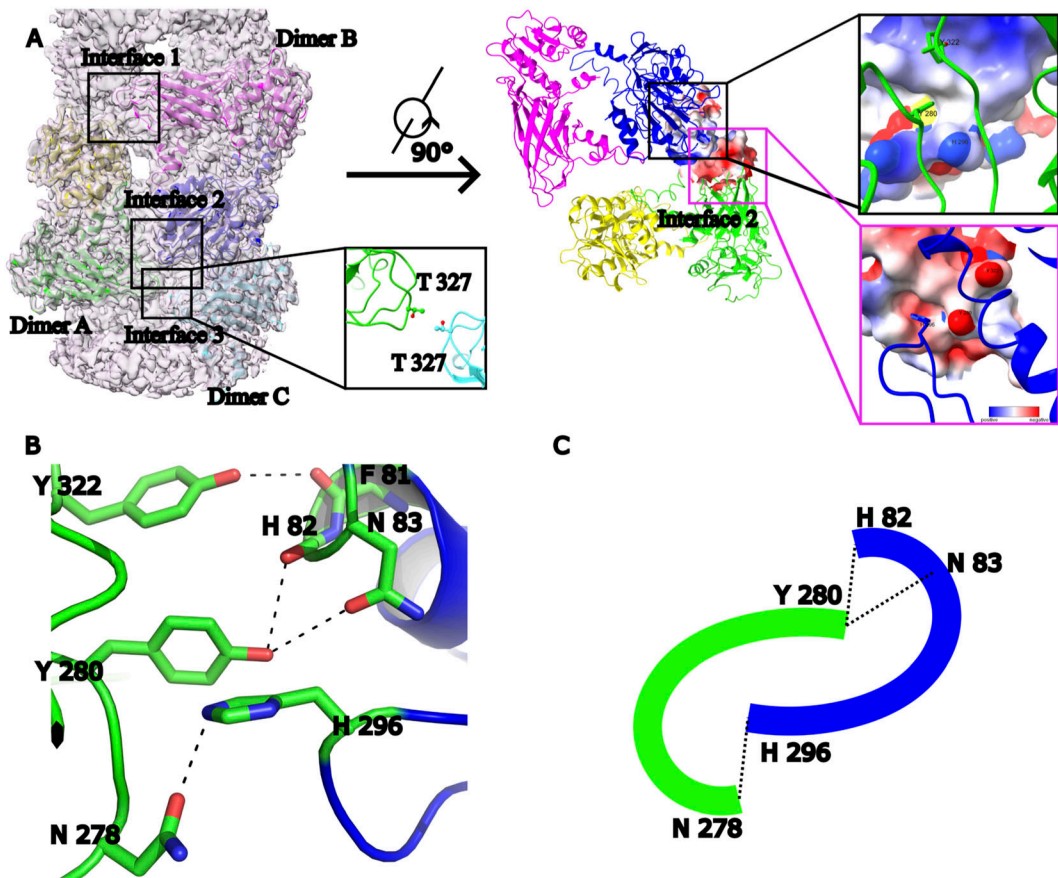

**Figure 3. The molecular mechanism of PRMT1 helical polymer.**
**(A)** Three dimers connect to form the helical polymer (Dimers A, B, and C). Two identical interfaces (interfaces 1 and 2) between dimers A and B and a weak contact between dimers A and C (interface 3) are shown. T327–T327 on interface 3. Y280 on the loop from dimer A (green) is inserted into the electrostatic pocket of dimer B (blue). H296 on the loop from dimer B is inserted into the electrostatic pocket of dimer A. **(B)** The contact in the interface contains three hydrogen bonds, which are N278 and H296, Y280 and H82 (or N 83), Y322 and F81. **(C)** These contacts form an interesting pattern in two "U-shape" pockets inserted into each other face-to-face.

supporting the notion that PRMT1 uses its oligomeric form, rather than its dimeric form, to bind RGG-rich substrates. However, we were unable to resolve the structural details of PRMT1 recognition of RGG regions, largely because of the inherent disorder of RGG domains.

To further highlight the role of helical polymers in facilitating PRMT1 binding to RGG-rich substrates, we systematically compared the binding affinities of various PRMT family members for these substrates. PRMT1 and PRMT8 displayed notably higher affinities for RGG-rich substrates compared with other PRMT family members, including PRMT2, PRMT3, PRMT5, and PRMT7 (Fig S4G–J). Notably, the previously reported crystal structure of PRMT8 indicates its assembly into helical filaments (45), which resembles the helical polymer structure of PRMT1 obtained in this study. This finding further emphasizes the importance of the helical polymeric scaffold for PRMTs in their interaction with RGG-rich substrates.

### Oligomerization enhances the enzymatic activity of PRMT1 toward substrates containing RGG motifs

The binding data presented above raises the question of whether PRMT1 oligomerization enhances its enzymatic activity toward RGG-rich substrates by improving substrate binding. To further explore the relationship between the oligomeric state and the activity of PRMT1, we measured the methyltransferase activity of WT PRMT1$^{oligomer}$ and mutant PRMT1$^{dimer}$ against various RGG-rich substrates. Given that dimerization is crucial for AdoMet binding in PRMTs and that the PRMT1$^{monomer}$ nearly loses its enzymatic activity, we will not further compare the PRMT1$^{monomer}$. The activities of PRMT1$^{oligomer}$ and PRMT1$^{dimer}$ were assessed across a range of concentrations (0.2 to 12.8 µM) with saturating concentrations of histone H4, hnRNPA1, and fibrillarin as substrates, respectively, in vitro. Unexpectedly, we found that PRMT1$^{oligomer}$ and PRMT1$^{dimer}$ exhibited nearly identical activity with histone H4, hnRNPA1, and fibrillarin under our experimental conditions (Fig S5A–D). These results clearly demonstrate that both the oligomeric and dimeric forms of PRMT1 are enzymatically active in vitro, supporting the notion that the dimer represents the minimal unit required for PRMT1 methyltransferase activity.

Although there were no marked differences in enzymatic activity between the oligomeric and dimeric forms of PRMT1 toward RGG-rich substrates in vitro, we observed that the assay conditions described above are much simpler than the complex and crowded

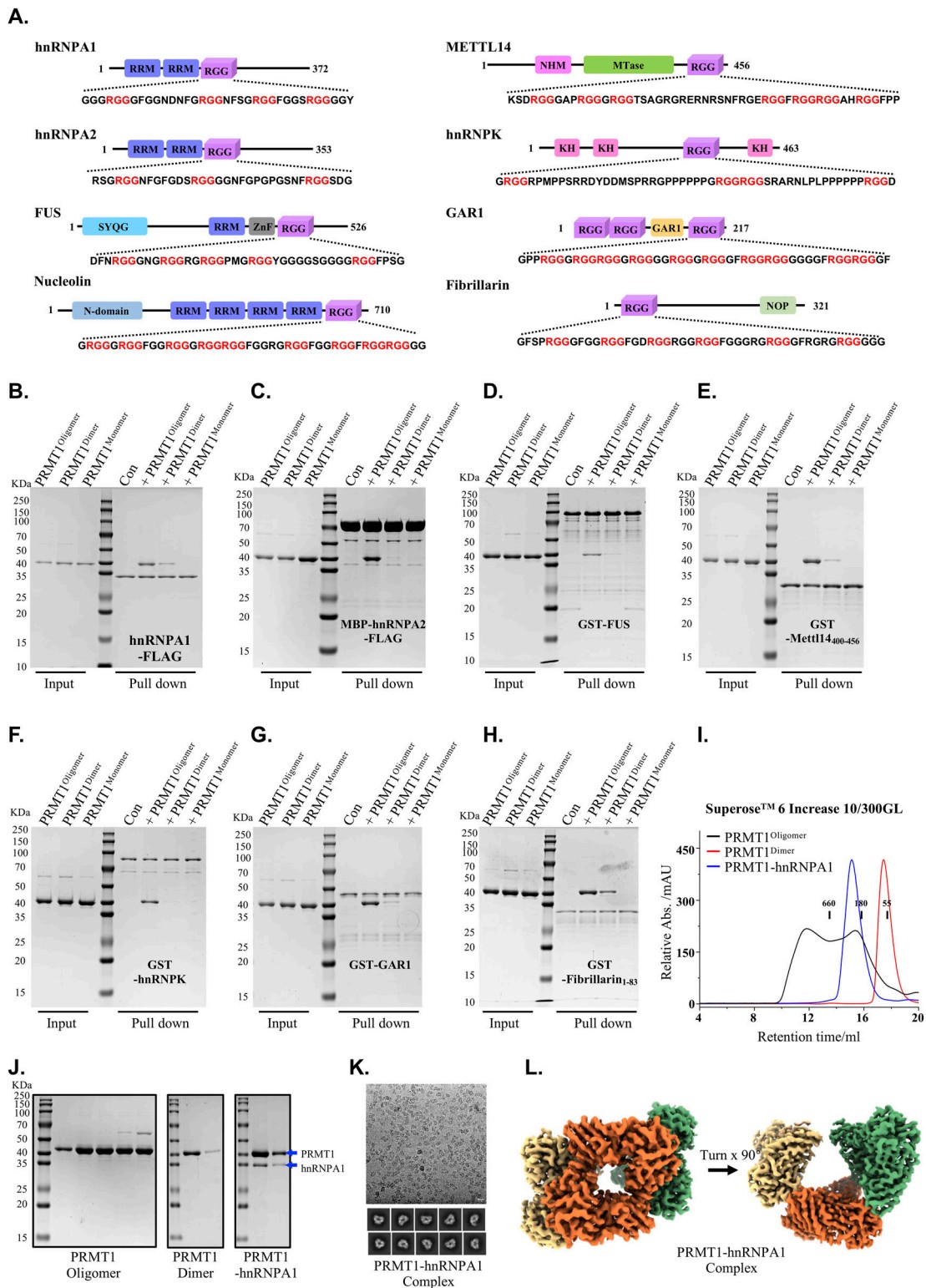

**Figure 4. The oligomerization of PRMT1 promotes its binding to substrate proteins enriched with RGG motifs.**
**(A)** The domain structure of RGG-rich substrate proteins, including hnRNPA1, hnRNPA2, FUS, Nucleolin, METTL14, hnRNPK, GAR1, and Fibrillarin, as well as the amino acid sequences of their RGG regions, are presented. **(B, C, D, E, F, G, H)** In vitro pull-down experiments were conducted to compare the binding affinities of various proteins, including hnRNPA1-FLAG, MBP-hnRNPA2-FLAG, GST-FUS, GST-METTL14 (400–456), GST-hnRNPK, GST-GAR1, and GST-Fibrillarin (1-83), with different oligomeric states of PRMT1, such as WT oligomers, Y280A/H296A/T327A mutant dimers, and W215A/Y220A/F222A mutant monomers. **(I, J)** The size exclusion chromatography elution profile of the PRMT1-hnRNPA1 complex was analyzed. The PRMT1-hnRNPA1 complex, along with WT PRMT1 oligomers and Y280A/H296A/T327A mutant PRMT1 dimers, was

intracellular environment. Macromolecular crowding in the intracellular milieu is known to influence various protein properties, including protein-protein interactions (46). We hypothesized that the oligomeric form of PRMT1, because of its higher affinity for RGG-rich substrates, may more effectively overcome binding resistance, thereby promoting the formation of asymmetric dimethylarginine (ADMA) on these substrates in vivo. To test this hypothesis, we compared the different oligomeric states of PRMT1 in catalyzing RGG-rich substrates in HEK293T cells. A series of FLAG-tagged RGG-rich substrates was co-transfected with distinct oligomeric states of MYC-PRMT1 into HEK293T cells. The overexpressed RGG motif substrates, catalyzed by the various oligomeric forms of PRMT1 in living cells, were captured using anti-FLAG resin. After co-immunoprecipitation, the specific binding of PRMT1 oligomeric states, along with the ADMA-modified RGG substrates, was released by FLAG peptide and visualized by Western blotting. In contrast to the in vitro enzymatic assay results, the oligomeric forms of PRMT1 substantially enhanced enzymatic activity toward RGG-rich substrates (Fig 5A–H). This finding is consistent with the observation that the binding of RGG motif substrates is substantially higher for PRMT1 oligomers compared with PRMT1 monomers and dimers in HEK293T cells, further underscoring the crucial role of PRMT1 oligomerization in binding RGG motif substrates to promote ADMA modification in living cells (Fig 5A–H). To specifically assess the ADMA levels attributable to endogenous PRMT1 activity, we introduced a critical negative control for each condition—Flag immunoprecipitation of each RGG motif-containing protein co-expressed with an empty MYC vector. Our results showed that ADMA signals under the empty vector conditions were minimal compared with those observed with PRMT1 overexpression. These findings indicate that the differential methylation patterns are primarily driven by the distinct oligomeric states of exogenously expressed PRMT1, with only a negligible contribution from endogenous PRMT1 (Fig 5A–H).

### Loss of PRMT1 oligomerization impairs global ADMA levels and suppresses PDAC tumor growth

To investigate how PRMT1 oligomerization affects its biological functions in vivo, it is essential to determine whether higher-order oligomerization occurs under physiological conditions. We selected human PDAC cells based on previous studies that identified PRMT1 as a critical factor for PDAC maintenance (14, 15, 16, 32). In addition, we used HeLa cells because of their widespread application in transfection experiments. To exclude the possibility that PRMT1 oligomer formation in living cells results from overexpression, we used CRISPR-mediated genome editing to knockout endogenous PRMT1 in HeLa cells (Fig S6A and B). Distinct oligomeric states of PRMT1-expressing HeLa cells were generated from the PRMT1 knockout (KO) HeLa cell line. These HeLa cells expressed comparable amounts of PRMT1 oligomers, dimers, and monomers to the

endogenous PRMT1 for further investigation (Fig S6C). Similarly, different oligomeric states of PRMT1-expressing PDAC cells were established from PRMT1-depleted human pancreatic cancer cell lines, specifically PANC-1 (Fig 6A and B).

Cell lysates from HeLa cells harboring different oligomeric states of PRMT1 were prepared and analyzed using SEC followed by Western blotting. In untreated HeLa cells, we observed that endogenous human PRMT1 was present at molecular weights exceeding ~200 kD (Fig S6D). A similar elution profile was observed for PRMT1 derived from HeLa cells expressing PRMT1$^{oligomers}$, confirming that the rescue of PRMT1 knockout (KO) HeLa cells with exogenous WT PRMT1 preserved an oligomeric state comparable with that of endogenous PRMT1 (Fig S6D). In contrast, PRMT1 from HeLa cells expressing PRMT1 dimers and monomers primarily eluted in a narrow peak corresponding to the dimeric or monomeric state of PRMT1 (Fig S6D). Similar results were also observed in PANC-1 cell lines (Fig 6C), suggesting that endogenous human PRMT1 exists as a higher-order oligomer in both HeLa cells and the human pancreatic cancer cell line PANC-1.

We subsequently investigated the correlation between the oligomeric states of PRMT1 and its methyltransferase activity in vivo. Recent studies have shown that global inhibition of asymmetric arginine methylation suppresses PDAC tumor growth (14). Given that oligomerization enhances PRMT1 activity by increasing its binding affinity for RGG-rich substrates, we aimed to determine whether the oligomeric state of PRMT1 is required for the growth of PDAC cells. To validate this hypothesis, we engineered PANC-1 cells using two independent shRNAs targeting PRMT1 or a non-targeting (NT) control shRNA (Fig 6D). We observed a reduction in ADMA levels and an accumulation of MMA in PRMT1-depleted PANC-1 cells compared with NT shRNA controls (Fig 6D). Similar results were also observed in MiaPaca-2 (Fig S6E). Ectopic overexpression of a shRNA-resistant PRMT1 oligomer, but not PRMT1 monomer cDNA, restored ADMA and MMA to physiological levels, with partial restoration observed upon expression of the PRMT1 dimer isoform, underscoring the crucial role of the PRMT1 oligomeric state in sustaining its methyltransferase activity in both PDAC cell lines (Figs 6E and S7A). In addition, depletion of PRMT1 significantly inhibited cell proliferation and colony formation in both PDAC cell lines, PANC-1 (Fig 6F and G) and MiaPaca-2 (Fig S6F and G), indicating a strong dependence of PDAC cells on PRMT1. Moreover, ectopic overexpression of an shRNA-resistant PRMT1$^{oligomer}$ but not PRMT1$^{dimer}$ or PRMT1$^{monomer}$ cDNA, rescued the growth defect in PRMT1-depleted PDAC cells (Figs 6H and I and S7B and C). Collectively, these data imply that the oligomer, rather than the dimer, is sufficient to promote the growth of PDAC cells such as PANC-1 and MiaPaca-2.

Ultimately, we investigated whether the oligomeric state of PRMT1 is required for maintaining PDAC tumor growth in vivo. PRMT1-depleted PANC-1 cells, rescued with various oligomeric

---

examined using Superose 6 Increase 10/300GL. In this analysis, PRMT1 oligomers and PRMT1 dimers served as controls, and the corresponding gel filtration fraction peaks were collected and analyzed via SDS–PAGE. **(K)** The cryo-EM micrograph and the 2D classification results of the PRMT1-hnRNPA1 complex are presented. **(L)** The 3D reconstruction of the PRMT1-hnRNPA1 complex reveals that it is a hexamer. In this figure, each color represents one PRMT1 dimer structure. Three dimers (colored yellow, orange, and green) assemble to form a hexameric structure.
Source data are available for this figure.

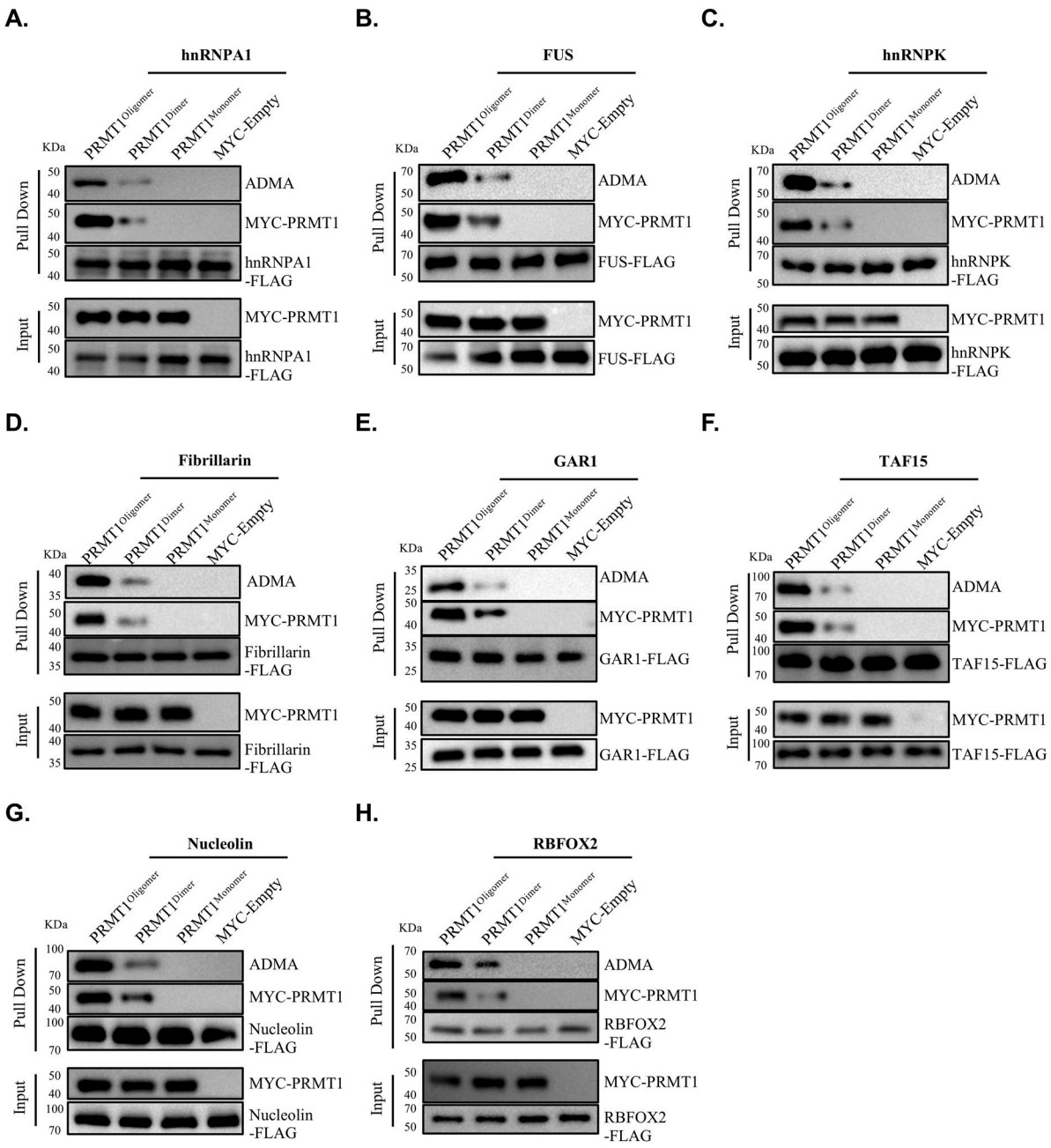

**Figure 5. The oligomerization of PRMT1 enhances its binding to and catalysis of substrate proteins enriched with RGG motifs in living cells.**
**(A, B, C, D, E, F, G, H)** Co-immunoprecipitation (Co-IP) assays were performed in HEK293T cells co-transfected with FLAG-tagged RGG motif-containing substrate proteins—hnRNPA1 (A), FUS (B), hnRNPK (C), Fibrillarin (D), GAR1 (E), TAF15 (F), Nucleolin (G), and RBFOX2 (H)—along with different MYC-tagged PRMT1 constructs: WT (oligomer), Y280A/H296A/T327A mutant (dimer), W215A/Y220A/F222A mutant (monomer), or an empty MYC vector as control. After FLAG affinity purification, eluates were analyzed by immunoblotting with anti-MYC and anti-ADMA antibodies. Anti-MYC detects PRMT1-substrate interactions, although anti-ADMA reflects the methylation activity of each PRMT1 form. The ADMA signal in each panel corresponds to the molecular weight of the respective substrate protein. The MYC-empty control accounts for endogenous PRMT1 activity.
Source data are available for this figure.

forms, were transplanted into immunocompromised mice, and tumor growth was monitored after tumor establishment. Substantial tumor growth inhibition was observed over 21 d in mice with tumors derived from PRMT1-depleted PANC-1 cells compared to control groups (Fig 6J and K). This growth inhibition could be reversed by reintroducing PRMT1 oligomers, but not PRMT1 dimers or monomers (Fig 6J and K). These findings suggest that WT oligomerization, rather than dimerization, is sufficient to sustain PDAC tumor growth in vivo. Immunofluorescence staining of tumor tissues indicated that the expression levels of the different oligomeric

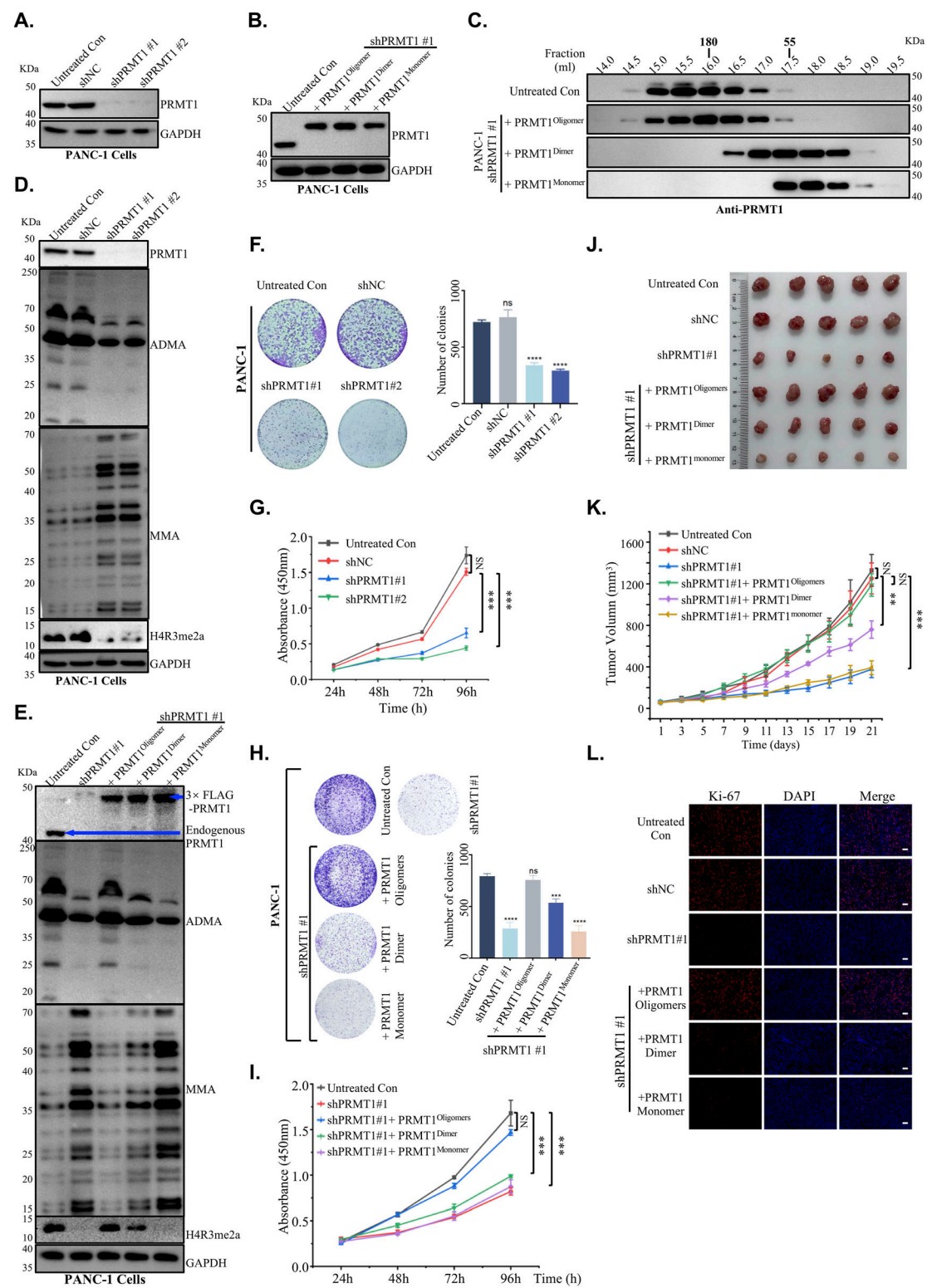

**Figure 6. Disruption of PRMT1 oligomerization impairs tumor growth.**
**(A)** PANC-1 cells were engineered using two independent PRMT1-targeting shRNAs (shPRMT1#1 and shPRMT1#2) and a non-targeting shRNA (shNC) to generate the corresponding cell lines. Western blot analysis was then performed to assess PRMT1 expression levels, with GAPDH serving as the loading control. **(B)** PANC-1 shPRMT1#1 cells were transiently transfected with shRNA-resistant PRMT1 oligomers, dimers, and monomers to attain PRMT1 expression levels comparable with those of untreated PANC-1 cells. The results were visualized using Western blot analysis, with GAPDH serving as the loading control. **(B, C)** Cell lysates from PANC-1 cells expressing different oligomeric states of PRMT1 and untreated PANC-1 cells produced in (B), underwent gel filtration chromatography to verify the endogenous oligomeric state of PRMT1 in

forms of PRMT1 were comparable with endogenous PRMT1 in the control group (Fig S7D). In addition, immunofluorescence staining for the cell proliferation marker KI-67, along with hematoxylin and eosin (HE) staining, confirmed that the loss of PRMT1 clearly inhibits tumor growth compared with the control group (Figs 6L and S7E). The reintroduction of PRMT1 oligomers, but not dimers or mono-mers, reversed tumor growth inhibition (Figs 6L and S7E). Finally, we performed Western blot analysis to assess ADMA/MMA levels in the collected tumors, aiming to confirm whether the rescue of tumor growth is because of the catalytic activity of PRMT1. Our Western blot results support the conclusion that the observed phenotype rescue, where WT oligomeric PRMT1—rather than dimeric or mo-nomeric forms—promotes PDAC tumor growth, is dependent on PRMT1's catalytic activity (Fig S7F). Collectively, these results un-derscore the critical dependence of established human PDAC xe-nografts on the oligomeric state of PRMT1 in vivo.

### Disruption of PRMT1 oligomerization inhibits PDAC tumor growth through affecting RNA metabolism

Among the numerous known RBPs, those containing RGG motifs rank as the second most prevalent RBPs in the human genome (26, 47, 48). RGG motif proteins play critical roles in regulating various facets of RNA processing, including mRNA transport, splicing, and translational regulation (48). RBPs that possess RGG/RG regions can form high-affinity interactions with RNA (49, 50, 51). Methylation of RGG/RG regions by PRMT1 may influence these interactions, thereby affecting RNA metabolism (52). PRMT1 has been identified as a critical factor in the maintenance of PDAC because of its fundamental role in RNA metabolism (14, 32). Given that the olig-omeric state is essential for PRMT1 to achieve stable binding to RGG-rich substrates and facilitate their methylation in vivo, we investigated whether perturbations in PRMT1 oligomerization, which we have observed to inhibit PDAC tumor growth, are closely linked to alterations in the functions of a subset of RBPs containing the RGG motif.

To explore the potential correlation between the disruption of PRMT1 oligomerization and alterations in RNA metabolism, we used a mass spectrometry-based proteomics approach to identify the differential substrates recognized by various oligomeric states of PRMT1 in PANC-1 cells (Fig 7A). The distinct oligomeric states of PRMT1 were enriched from PANC-1 cells through immunoaffinity purification using Anti-FLAG resin and subsequently identified via liquid chromatography-tandem mass spectrometry (LC-MS/MS) (Fig 7A). Analysis of the proteomics results revealed marked

changes in peptide abundance among the different oligomeric states of PRMT1, indicating oligomeric state-regulated binding of PRMT1 to substrates (Fig 7B). To gain further insights into the cellular functions of these substrates, which are differentially recognized by PRMT1 in its various oligomeric states, we performed gene ontology (GO) analysis by mapping the peptides back to their original proteins. We identified 36 proteins of a total of 619 that were differentially enriched in the oligomeric form of PRMT1 compared with its dimer and monomer forms (Fig 7B–D). These differentially enriched substrates were primarily associated with nine biological processes in the GO analysis (q < 0.05), including RNA stabilization, RNA splicing, and regulation of the mRNA cata-bolic process, with RNA stabilization being the most strongly enriched category (q < 0.001; 15 of 36 RNA stabilization genes enriched) (Fig 7E). Many differentially bound proteins were RBPs, including multiple heterogeneous nuclear ribonucleoproteins (hnRNPs) and other RGG motif-containing proteins involved in RNA metabolism (Fig 7C). This finding is consistent with our in vitro binding results and enhances the confidence in our experimental findings (Figs 4B–H and 5A–H).

Next, we used various oligomeric states of PRMT1-expressing PANC-1 cells, alongside untreated PANC-1 cells as a positive control and PRMT1-depleted PANC-1 cells as a negative control, to perform RNA sequencing (RNA-Seq) analysis. To identify the genes regulated by PRMT1, we used the Venn tool, which identified 674 genes that were considerably altered by PRMT1 depletion and could be res-cued by the re-expression of the oligomeric state of PRMT1 in PRMT1-depleted PANC-1 cells (Fig 7F). Subsequently, we compared the transcriptomes of these 674 genes in PANC-1 cells harboring different oligomeric states of PRMT1, including oligomeric, di-meric, and monomeric states, ultimately identifying 112 genes closely correlated with the different oligomeric states of PRMT1, as shown in the heat map (Fig 7G). Gene enrichment analysis of these 112 related genes indicated that they were associated with RNA-related signaling pathways, including "mRNA processing," "RNA splicing," and "regulation of mRNA metabolic processes," which align closely with the GO analysis derived from the aforementioned LC-MS/MS results (Fig 7E and H). To validate our RNA-seq data, we randomly selected seven genes for RT-qPCR analysis. The RT-qPCR results showed a high level of concor-dance with the RNA-seq data (Fig S8A–G). Collectively, these results further reinforce the necessity of the oligomeric state for PRMT1's functionality in PDAC and confirm that the disruption of PRMT1 oligomerization suppresses tumor growth by influencing RNA metabolism.

---

PANC-1. Visualization was performed using Western blot analysis. **(D)** Western blot analysis was conducted to detect the levels of ADMA, MMA, and H4R3me2a in PANC-1 cells after the depletion of PRMT1. GAPDH was included as the loading control. **(E)** Re-expression of oligomeric PRMT1, dimeric PRMT1, or monomeric PRMT1 in PANC-1 cells after PRMT1 depletion was conducted, and Western blot analysis was performed to measure the levels of ADMA, MMA, and H4R3me2a in the cells, using GAPDH as the loading control. **(F, G)** Colony formation and proliferation assays were conducted in PANC-1 cells as follows: untreated PANC-1 cells, two independent PRMT1-targeting shRNAs (shPRMT1#1 and shPRMT1#2), and a non-targeting shRNA (shNC), followed by analysis using one-way ANOVA. **(H, I)** Colony formation and cell proliferation assay results of PANC-1 cells expressing different oligomeric states of PRMT1, along with untreated PANC-1 cells and PRMT1-depleted PANC-1 cells serving as controls, were analyzed using one-way ANOVA. **(J)** PRMT1 knockdown PANC-1 cells (shPRMT1#1) were stably transfected to express oligomeric, dimeric, or monomeric forms of PRMT1 at levels comparable with endogenous PRMT1 in untreated PANC-1 cells. These reconstituted cell lines, along with PANC-1 cells expressing non-targeting shRNA (shNC) as a negative control, were subcutaneously injected into female BALB/c nude mice (n = 5 per group) for xenograft tumor formation. Tumors were harvested after 21 d. **(J, K)** Tumor growth curves corresponding to the xenografts in (J) were measured over time and analyzed using two-way ANOVA. **(J, L)** Immunofluorescence staining for Ki-67 was performed on tumor sections obtained from (J) to evaluate cellular proliferation. Scale bar: 50 $\mu$m.
Source data are available for this figure.

**Figure 7.   The oligomerization of PRMT1 promotes the growth of PDAC by influencing the cascade of RNA-binding proteins.**
**(A)** A flowchart illustrating the experimental procedure for identifying the differential cellular substrates recognized by various oligomeric states of PRMT1 in PANC-1 cells using a mass spectrometry-based proteomics approach is provided. **(A, B)** A Venn diagram illustrating the high-affinity substrates associated with PRMT1 oligomers, dimers, and monomers from the experiment described in (A) is presented. **(B, C)** Among the 66 substrate proteins identified in the overlap between substrates enriched from PRMT1 oligomers and dimers derived from the Venn diagram in (B), 36 proteins were selected based on an abundance ratio (Oligomers/Dimers) greater than or equal to two, indicating substantial differences in binding between the oligomeric and dimeric states of PRMT1. **(C, D)** Abundance ratios of the differential proteins bound by dimers and monomers, as presented in (C). **(C, D, E)** GO biological process enrichment analysis of the substrates with varying affinities from (C, D). **(F)** A Venn diagram illustrating the overlap between PRMT1 knockdown and the re-expression of PRMT1 oligomers in PRMT1 knockdown cells. **(G)** A heatmap displaying gene expression

# Discussion

Several studies have demonstrated that PRMT1 forms oligomers larger than dimers in vitro through the use of SEC, analytical ultracentrifugation (AUC), and chemical crosslinking (21, 24, 45, 53, 54, 55, 56). However, the biological function of PRMT1 oligomers in vivo has not yet been fully elucidated. Dimerization is a conserved characteristic among all type I PRMTs and is essential for normal catalytic activity (18). Consequently, dimerization has long been regarded as necessary for the biological functions of PRMT1; yet, it remains uncertain whether intracellular dimerization alone suffices for PRMT1 to fulfill these functions and whether higher-order oligomerization occurs under physiological conditions. Few studies have systematically compared the biological functions of oligomers and dimers in vivo, largely because of a lack of structural data regarding the oligomeric state of human PRMT1. Although the crystal structure of rPRMT1 has been previously reported as a homo-dimer, higher-order oligomers were detected in solution (17), suggesting that crystallization conditions may have disrupted rPRMT1 oligomerization (17, 24). To date, a homo-oligomeric structure has been identified in *Saccharomyces cerevisiae* PRMT1 (ScPRMT), which forms a ring-like hexamer (22); however, ScPRMT1 shares only 50% sequence identity with hPRMT1. Thus, the structural basis for hPRMT1 oligomerization remains unresolved.

In this study, we determined the cryo-EM structure of human PRMT1 in its oligomeric form. Furthermore, we characterized hPRMT1 using a combination of biochemical assays, mutagenesis experiments, and functional assays. SEC revealed that PRMT1 elutes as a broad peak ranging from 200 to 600 kD, indicating the formation of heterogeneous oligomers. Negative staining further corroborated that PRMT1 assembles into a concentration-dependent helical polymer. We demonstrated that endogenous PRMT1 exists as oligomers larger than dimers under physiological conditions. Analysis of the Cryo-EM structure of hPRMT1 led to the identification of two sets of amino acid residues at different interfaces of the oligomers, which are involved in stabilizing the dimerization and oligomerization of PRMT1. In addition, we generated monomeric and dimeric PRMT1 mutants for functional studies. Finally, we revealed that the disruption of PRMT1 oligomers into dimers or monomers impairs its binding affinity for a series of RBPs containing RGG/RG motifs, thereby reducing arginine methylation levels on these substrates in living cells.

Proteins containing RGG motifs rank as the second most prevalent group of RBPs (26, 47, 48). These proteins have been shown to undergo liquid-liquid phase separation both in vitro and in vivo (29, 57, 58), a process likely regulated by various post-translational modifications, including phosphorylation and methylation (59). Several studies have demonstrated that PRMT1 methylates RGG substrates to modulate their phase behavior (7, 60). In this study, we found that the oligomeric state of PRMT1 is required to form a stable complex primarily in a hexameric form with RGG motif substrates in a homogeneous hexameric form on an SEC column, at least for

hnRNPA1 and fibrillarin that we tested. Given that phase separation is highly dependent on the degree of protein oligomerization, our findings provide additional insight into how PRMT1 may regulate the phase separation of RBPs containing RGG motifs beyond its previously established enzymatic activity. This adds another layer of regulatory complexity to the process of phase separation of RGG motif-containing RBPs regulated by PRMT1. Further investigation is needed to determine the extent to which the phase separation of RGG substrates depends on PRMT1's enzymatic activity versus their direct binding. In the future, the application of separation-of-function PRMT1 mutations, based on the cryo-EM structure of hPRMT1 reported here, will enable us to gain a more comprehensive understanding of this topic.

In this work, we screened and identified that both PRMT1 and PRMT8 exhibit much higher affinity for substrates enriched with RGG motifs compared with other members of the PRMT family. Structural analyses revealed that both the previously reported crystal structure of PRMT8 and the cryo-EM structure of PRMT1, presented in this study, adopt a similar helical polymer conformation. This finding strongly suggests that the helical polymer structural scaffold may represent a predominant conformation for PRMTs when recognizing substrates containing the RGG motif. Our results indicate that the helical polymer may account for PRMT1's inherent preference for RGG motifs in vivo and in vitro. To gain further insights into their interaction details, we attempted to resolve the cryo-EM structures of the PRMT1-hnRNPA1 and PRMT1-fibrillarin complexes; however, we were unable to elucidate the structural details of hnRNPA1 or fibrillarin within these complexes because of inherent disorder present in the RGG regions.

The regulation of RNA metabolism by PRMT1 has been demonstrated as a primary mechanism for sustaining PDAC (14, 32). In this study, we showed that the depletion of PRMT1 significantly inhibits the proliferation of pancreatic cancer cells, including PANC-1 and MiaPaca-2. Notably, the restoration of PRMT1's oligomeric forms, but not its dimeric or monomeric forms, can reverse this inhibition, indicating that the oligomeric state of PRMT1 is essential for sustaining pancreatic cancer cell proliferation. To elucidate the mechanisms by which PRMT1 supports this proliferation, we established various oligomeric states of PRMT1 expression in PANC-1 cells and used a mass spectrometry-based proteomics approach to identify the differential substrates recognized by the oligomeric forms of PRMT1. We conducted GO analysis of proteins enriched in the oligomeric form of PRMT1 compared with its dimeric and monomeric forms, revealing that the differentially enriched substrates are RBPs rich in RGG motifs, primarily associated with RNA processing. Therefore, the findings of this study suggest that the disruption of PRMT1 oligomerization markedly inhibits the proliferation of pancreatic cancer cells, likely because of disturbance of PRMT1's binding to RBPs enriched with RGG motifs.

The RGG region in RBPs serves as a preferential site for arginine methylation by PRMT1, a process that regulates their binding affinity to RNA. Consequently, alterations in the methylation status of RBPs catalyzed by PRMT1 can affect aberrant mRNA biogenesis. Our

---

levels affected by the different oligomeric states of PRMT1. **(G, H)** GO biological process enrichment analysis of the genes identified in (G) was performed using Hiplot Pro (https://hiplot.com.cn).

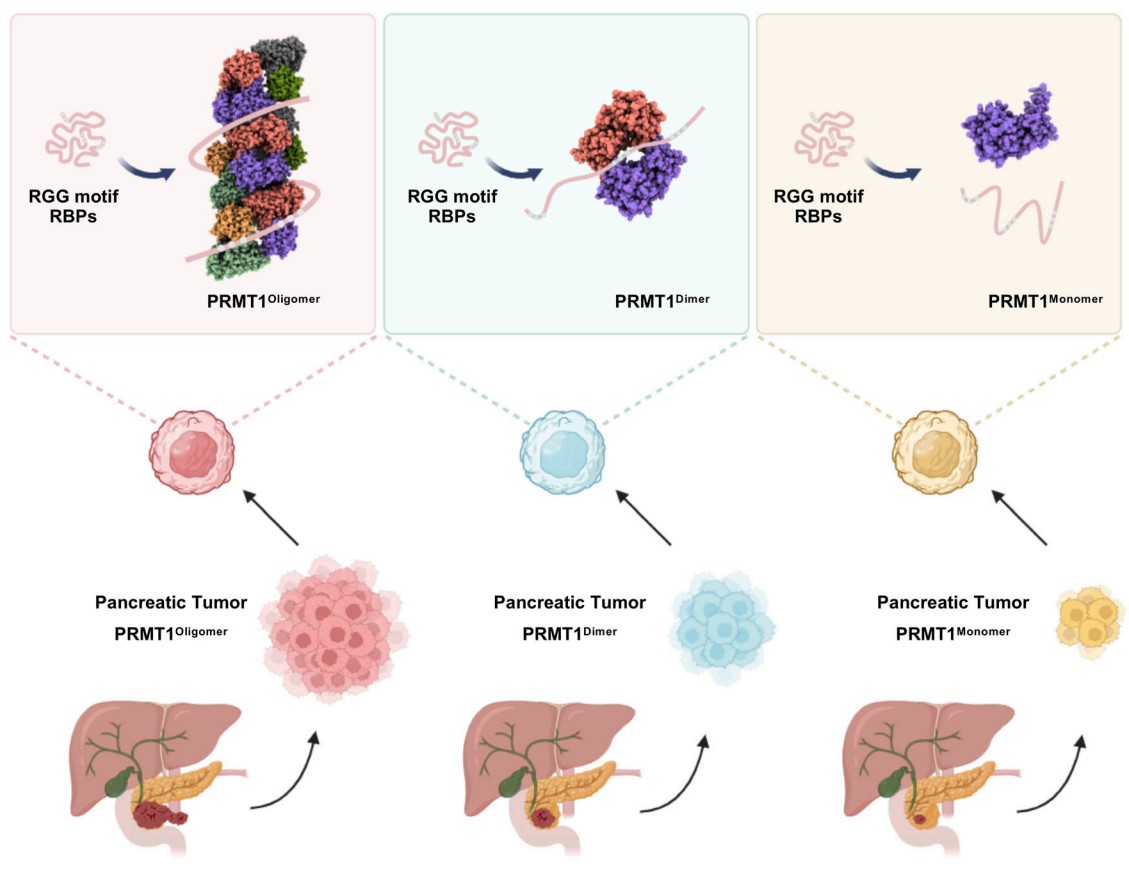

RBPs: RNA-binding proteins

RGG motif: Arginine/glycine-rich region

**Figure 8. Proposed model illustrating the role of oligomerization-mediated PRMT1 activity in the progression of pancreatic ductal adenocarcinoma (PDAC).**
The oligomeric state of PRMT1 enhances its affinity for RNA-binding proteins enriched in RGG motifs, thereby facilitating the methylation of arginine residues. Disruption of PRMT1 oligomerization—shifting it to dimeric or monomeric forms—impairs its interaction with these proteins, leading to a reduction in asymmetric arginine methylation. This, in turn, disrupts key RNA metabolic processes, including RNA splicing and mRNA processing, ultimately suppressing tumor growth in PDAC. Collectively, these findings support the hypothesis that PRMT1 oligomerization is critical for PDAC progression and highlight it as a promising therapeutic target.

transcriptomic data suggest that the disruption of PRMT1 oligomerization leads to substantial changes in co-transcriptional RNA processing, including splicing alterations. Thus, in combination with the previously described mass spectrometry-based proteomics data, disruption of PRMT1 oligomerization decreases its binding to several RBPs, consequently impairing the methylation of arginine residues in the RGG regions of these substrates. The combined transcriptomic and proteomics data support the conclusion that the loss of PRMT1 oligomerization inhibits PDAC growth by affecting the arginine methylation of RBPs, thereby disrupting RNA metabolism. Collectively, our study proposes a model wherein PRMT1 oligomerization enhances its binding to RGG motif-containing RBPs, promoting arginine methylation. Disruption of PRMT1 oligomerization into dimeric/monomeric forms impairs these interactions, reducing asymmetric arginine methylation and consequently disrupting critical RNA metabolic processes. This cascade ultimately suppresses PDAC tumor growth, establishing PRMT1 oligomerization

as both a critical driver of PDAC progression and a promising therapeutic target (Fig 8).

During the preparation of our manuscript, a recent study by Dang et al similarly reported the cryo-EM structure of human PRMT1 oligomers, including tetramers, hexamers, and helical filaments, corroborating our observation that PRMT1 is capable of forming higher-order assemblies (61). Their findings further support our conclusion that oligomerization enhances PRMT1's enzymatic activity. Although complementary, our study diverges in several key aspects. First, we systematically dissected the functional roles of distinct oligomeric states by generating interface-specific mutants (e.g., PRMT1[Y280A/H296A/T327A] for dimer; PRMT1[W215A/Y220A/F222A] for monomer). This approach enabled us to directly evaluate how oligomerization and dimerization differentially influence substrate recognition, particularly for RGG motif-containing RBPs, which are key physiological substrates of PRMT1. Second, unlike the reference 61, who used RGG motif-containing peptides, our study focused on

full-length, physiologically relevant RBPs. We found that oligomeric PRMT1 facilitates multivalent substrate engagement, notably enhancing binding affinity and catalytic efficiency under native cellular conditions. Third, we provided direct evidence that PRMT1 predominantly exists in oligomeric forms in vivo. Using CRISPR-edited HeLa cells expressing PRMT1 variants at endogenous levels, SEC revealed that native PRMT1 primarily assembles into higher-order oligomers, reinforcing the physiological relevance of our structural findings. Finally, we demonstrated the pathophysiological importance of PRMT1 oligomerization in PDAC. Only the oligomeric form was capable of rescuing tumor growth in PRMT1-depleted PDAC models, directly linking PRMT1's structural state to its tumor-promoting function and highlighting the therapeutic potential of targeting its oligomeric assembly.

# Materials and Methods

### Cell culture

HEK293T and HeLa cells were obtained from the American Type Culture Collection (ATCC), although PANC-1 and MIA PaCa-2 cells were purchased from Servicebio, China. HEK293T, HeLa, and MiaPaCa-2 cells were grown in DMEM supplemented with 10% (vol/vol) FBS and 1% penicillin/streptomycin at 37°C in a 5% $CO_2$ atmosphere. In contrast, PANC-1 cells were cultured in RPMI 1640 medium, also supplemented with 10% (vol/vol) FBS and 1% penicillin/streptomycin, under the same conditions of 37°C and 5% $CO_2$.

### Animal studies

Female BALB/c nude mice (5–6 wk old) were purchased from GemPharmatech, China, and maintained in the specific pathogen-free (SPF) facility of Xiamen University's Animal Laboratory. The mice were housed on a 12-h light/dark cycle with unrestricted access to standard chow and water. All animal experiments received approval from the Xiamen Animal Care and Use Committee.

### Protein purification

The cDNA sequences encoding human PRMT1 (residues 32–371) and its mutant proteins were subcloned into a pET-28a vector containing an N-terminal His6-Sumo-StrepI tag. A TEV protease cleavage site was inserted between the Sumo-StrepI tag and PRMT1. Both WT and mutant His6-Sumo-StrepI-TEV-PRMT1 proteins were transformed into *E. coli* BL21 (DE3) cells and subsequently expressed. The cells were cultivated in LB medium at 37°C until an optical density (OD600) of 0.8 was reached, followed by induction with 0.5 mM IPTG at 20°C for 15 h. For protein purification, the cells were harvested and lysed by sonication in a lysis buffer containing 50 mM Tris–HCl (pH 8.0), 500 mM NaCl, 10% glycerol, and 5 mM 2-mercaptoethanol, supplemented with 1 mM PMSF. After centrifugation, the supernatant was applied to Ni-NTA resin, and the bound protein was extensively washed with Ni-NTA wash buffer (50 mM Tris, pH 8.0; 500 mM NaCl; 10% glycerol; 20 mM imidazole; 5 mM 2-

mercaptoethanol) and subsequently eluted using wash buffer supplemented with 250 mM imidazole. Strep-Tactin resin was subsequently used for further purification after the His6-Sumo tag was cleaved from StrepI-TEV-PRMT1 by SUMO protease. The protein eluted from the Strep-Tactin resin was pooled, concentrated, and loaded onto a Superdex 200 Increase 10/300GL SEC column (Cytiva) in a buffer containing 20 mM Tris–HCl (pH 8.0), 150 mM NaCl, and 2 mM DTT. The peak fractions were pooled, concentrated, flash-frozen in liquid nitrogen, and stored at −80°C.

### Negative stain sample preparation

WT PRMT1, purified via SEC, was prepared at concentrations of 0.1, 0.2, 0.4, and 0.6 mg/ml to investigate the concentration threshold of forming helical polymer. All various concentration samples were prepared following this protocol: First, 5 μl of protein samples were placed on a glow-discharged grid coated with continuous carbon film (Beijing Zhongjingkeyi Technology Co., Ltd.) and incubated for 90 s. Second, the excess protein solution was removed using filter paper. Next, the grid was stained with 5 μl of 0.75% (wt/vol) uranyl acetate for 60 s, after which the excess uranyl acetate was removed using filter paper. Finally, the processed samples were examined using a 120 kV electron microscope (FEI Company). All the following mutant PRMT1 samples were prepared at the concentration threshold and observed in negative-stain microscopy with the same protocol.

### Cryo-EM sample preparation

4 μl of 0.6 mg/ml PRMT1 solution was dripped onto the glow-discharged 300 mesh Quantifoil Cu R 1.3/1.2 grid that placed in the Vitrobot (Thermo Fisher Scientific). The Vitrobot chamber was preset at a temperature of 4°C and a humidity level of 100%, and the rest parameters were set as following to achieve the desired ice thickness: waiting time of 10 s, blotting time of 4 s, and blotting force of 1. Then, the grid was promptly plunged into liquid ethane that had been pre-cooled with liquid nitrogen.

### Cryo-EM data acquisition

All the data were acquired using the Titan Krios G3i 300 kV electron microscope in which a spherical aberration corrector, a Gatan continuum (1,069) EELS Energy filter and a Post-GIF Gatan K3 Summit direct electron detector are equipped. Program SerialEM (62) is responsible for data collection under super-resolution mode at magnification of 130,000x with pixel size 0.46 Å. The defocus values were set from 1.5 to 2.5 μm. Each movie has 32 dose-fractionated frames with a total electron dose of 50 e⁻/Å² and an exposure time of 2.09 s 1,748 movie stacks were collected.

### Data processing

A total of 1,748 movie stacks were acquired and were first processed by MotionCorr2 (33) was used to correct the beam-induced motion of aligning the individual frames, resulting in dose-weighted micrographs with a final pixel size of 0.92 Å. The subsequent data processing was carried out using RELION 3.1.0 (63) and laboratory-

made programs. Defocus values for each micrograph were estimated using CTFFIND4 ([64]). Subsequently, filaments were picked manually from the selected 485 micrographs and high signal-to-noise ratio (SNR) filament segments were kept through 2D (two-dimensional) classification. These high-quality 2D segment classes were used in the subsequent 3D reconstruction and refinement, with the initial model generated using the laboratory-made program mrcReference. In addition, we determined the initial helical parameters of helical rise of 19.23 Å and twist angle of 102.86° using our laboratory-made program called Picker from these data.

Automated filament picking was completed by combining the Topaz program ([65]), and our laboratory-made program helixPick. Shortly, all the filament segments were found by the Topaz program and the filament were identified by helixPick from the segment's coordinates. Then, Relion extracted all the filaments following the routine procedure. A total of 2,109,227 segments were extracted and filtered by 2D classification procedure. The selected segments were further used in the subsequent several rounds of Refine3D and three-dimensional (3D) classification and refinement. A final set of 150,914 segments were selected. The map was reconstructed at a resolution of 3.68 Å with the helical parameters, including a twist angle of 104.156 and a helical rise of 24.5843 Å.

### Analytical gel filtration

The SEC analysis of various oligomeric states of human PRMT1, specifically the WT PRMT1 oligomer, the Y280A/H296A/T327A mutant PRMT1 dimer, and the W215A/Y220A/F222A mutant PRMT1 monomer, were conducted using a Superdex 200 Increase 10/300GL column in SEC buffer (20 mM Tris, pH 8.0; 150 mM NaCl; 2 mM DTT). Thyroglobulin (660 kD), MLL1 complex (179 kD), and FTO (55 kD) served as calibration standards. Protein samples were injected at a flow rate of 0.35 ml/min, and the elution profile was monitored at 280 nm.

### PRMT1 oligomerization Co-IP assay

HEK293T cells (ATCC) were used in PRMT1 oligomerization assay. PRMT1 cDNA (WT or mutants) was subcloned into the pCDHEF1 vector with an N-terminal FLAG tag or Myc tag to generate FLAG-PRMT1 or Myc-PRMT1, respectively. Constructs for expressing WT or mutant FLAG-PRMT1 and Myc-PRMT1 were co-transfected into HEK293T cells. After 48 h, the cells were harvested, and the nuclear extracts and cytosolic fractions were prepared following standard protocols. The nuclear extracts and cytosolic fractions were captured using Anti-FLAG resin. Non-specifically bound species were removed by extensive washing with pull-down buffer (50 mM Tris–HCl, pH 8.0; 150 mM NaCl; 10% glycerol; 0.1% NP40; and 2 mM DTT), and bound proteins were eluted using FLAG peptide at 4°C. Anti-FLAG and anti-Myc signals were examined by Western blotting with the respective anti-epitope antibodies.

### GST/FLAG pull-down assay

Briefly, GST-tagged FUS, hnRNPK, GAR, Fibrillarin (1–83), and Mettl14 (400–456), or FLAG-tagged hnRNPA1 and hnRNPA2, were pre-incubated with GST beads or anti-FLAG resin, respectively, and then mixed with WT or mutant PRMT1 purified from *E. coli* in the

binding buffer (50 mM Tris–HCl, pH 8.0; 150 mM NaCl; 2 mM DTT; 10% glycerol; and 0.1% NP40) for 1 h at 4°C. After extensive washing with the same binding buffer, the supernatants were eluted using binding buffer containing GSH or FLAG peptide and were subsequently subjected to Coomassie staining.

### CRISPR-mediated PRMT1 knockout in hela cells

The PRMT1 knockout (KO) HeLa cell lines were generated using the CRISPR/Cas9 gene-editing system. A single-guide RNA (sgRNA) targeting sequence (5'-CGAGGCCGCGAACTGCATCA-3') was designed using an online tool (http://crispr.mit.edu) and cloned into the LentiCRISPR v2 vector (#52961; Addgene). HEK293T cells were co-transfected with the LentiCRISPR v2-PRMT1 (gRNA) vector, along with the packaging plasmid psPAX2 (#12260; Addgene) and the envelope plasmid pCMV-VSV-G (#8454; Addgene) using PEI reagent. After 48 h of transfection, the virus-containing medium was collected and used to infect HeLa cells. The PRMT1-KO HeLa cell line was established through selection with puromycin, and single-cell colonies were isolated. The knockout efficiency was assessed by Western blot analysis and PCR amplification of genomic DNA, followed by sequencing.

### Lentivirus production and delivery

Lentiviruses were packaged for pLKO.1-Hygro-shPRMT1#1, pLKO.1-Hygro-shPRMT1#2, and the control pLKO.1-Hygro-shNC using the packaging plasmid psPAX2 (#12260; Addgene) and the envelope plasmid pCMV-VSV-G (#8454; Addgene). Briefly, 2.25 µg of psPAX2, 0.75 µg of pCMV-VSV-G, and 3 µg of the corresponding constructs were co-transfected into HEK293T cells using PEI (#40815ES03; Yeasen Biotechnology). After 48 h of transfection, the virus-containing medium was collected and used to infect PANC-1 or MIA PaCa-2 cells at a 1:1 dilution. 48 h post-infection, 2 mg/ml of Hygromycin B was added to the culture to select for PRMT1 knockdown cell lines. PRMT1 Knockdown efficiency was assessed using Western blot analysis.

### Co-immunoprecipitation and arginine methylation assay in HEK293T cells

Briefly, 3×Flag-tagged full-length hnRNPA1, hnRNPA2, hnRNPK, METTL14, GAR1, FUS, Nucleolin, and Fibrillarin were cloned into the pCDHEF1 vector and co-transfected with various oligomeric states of PRMT1, specifically WT Myc-PRMT1[oligomers], mutant Myc-PRMT1[dimers], or Myc-PRMT1[monomers] into HEK293T cells. The nuclear and cytoplasmic extracts were prepared for anti-Flag resin binding for 2 h. The bound proteins were washed extensively with pull-down buffer (50 mM Tris, pH 8.0, 150 mM NaCl, 10% glycerol, and 0.1% NP40). Subsequently, the proteins were eluted using pull-down buffer containing Flag peptide for Western blot analysis.

### In vitro arginine methylation assays

For the PRMT1 enzymatic activity assay, various concentrations of purified WT PRMT1 or mutant PRMT1 were incubated with 0.125 µg of histone H4 and 32 µM SAM in 20 µl of methyltransferase assay buffer

(50 mM Tris [pH 8.0], 100 mM NaCl, 2.5 mM MgCl$_2$, 1 mM EDTA, and 2.5 mM DTT) at 37°C for 1 h. The reaction was terminated by adding 10 $\mu$l of 4 × SDS sample loading dye and boiling at 85°C for 5 min. Histone H4 asymmetrically dimethylated at arginine was separated by SDS–PAGE and detected by Western blotting using an anti-H4R3me2a primary antibody.

### Cell proliferation and colony formation assay

To investigate the role of various oligomeric states of PRMT1 in promoting pancreatic cancer cell growth, shPRMT1-resistant cDNA for PRMT1 was designed and subcloned into the pCDH-EF1-3×FLAG vector. In addition, site-specific mutations were introduced into this PRMT1 cDNA to produce both dimeric and monomeric forms of PRMT1. Subsequently, the different oligomeric states of pCDH-EF1-3×FLAG-PRMT1 (WT or mutant) were transfected into pancreatic cancer cell lines, including PANC-1 and MIA PaCa-2, which are deficient in PRMT1. This ensured that the expression levels of WT PRMT1 and its mutants were comparable with those of endogenous PRMT1. The established cell lines expressing distinct oligomeric states of PRMT1, along with untreated pancreatic cancer cells and PRMT1 knockdown pancreatic cancer cells, were used for cell proliferation and colony formation assays.

For the cell proliferation assays, the designated cell lines were seeded in 96-well plates at a density of 1,500 cells per well. Proliferation was assessed using the Cell Counting Kit-8. Briefly, the reagent solution was added to each well and incubated at 37°C for 1 h at various time points. Subsequently, cell proliferation was quantified by measuring absorbance at 490 nm, using wells containing only medium as blanks.

In the colony formation assay, cells were seeded in 35-mm dishes at a density of 1,000 cells per dish. After ~10 d of continuous culture, the cells were washed twice with PBS and subsequently stained with 0.05% crystal violet. The cells were then air-dried and photographed for further analysis.

### Tumor xenograft assay

The PRMT1-knockdown PANC-1 cell line was generated using the pLKO.1-Hygro-shPRMT1#1 vector. To establish PANC-1 cells that stably express the different oligomeric states of PRMT1, lentiviruses carrying either WT or mutant pCDH-EF1-Puro-3×Flag-PRMT1 (oligomer, dimer, and monomer) were packaged and used to infect the PRMT1-knockdown PANC-1 cell line. For the xenograft experiments, female BALB/c nude mice (aged 5–6 wk) served as recipients. The established cells were harvested, counted, and resuspended at a concentration of 1 million cells per 100 $\mu$l in PBS. This cell suspension was mixed in a 1:1 ratio with Matrigel (356234; Corning), and a total volume of 200 $\mu$l was injected subcutaneously into the right flank of the immunocompromised mice. The experimental groups included PANC-1, PANC-1-shPRMT1, and PANC-1-shPRMT1 rescue with PRMT1 oligomer, PRMT1 dimer, or PRMT1 monomer, all exhibiting expression levels comparable with endogenous PRMT1. Tumor size was measured every 2 d over a total period of 21 d. After the 21-d implantation period, the tumors were harvested, and their weights were recorded. Tumor growth was monitored using calipers, and tumor volume (TV) was calculated using the standard formula: (length × width (2))/2.

### Immunoprecipitation LC-MS/MS assay

Different oligomeric states of PRMT1 cDNA were subcloned into pCDH-EF1-Puro-3×Flag-PRMT1, along with the pCDH-EF1-Puro-3×Flag-empty vector, for lentiviral packaging to infect PRMT1-depleted PANC-1 cell lines. The resulting cell lines were grown in 15 cm dishes and lysed using a cell lysis buffer containing 10 mM Tris–HCl (pH 7.5), 150 mM NaCl, 1% Triton X-100, 5 mM EDTA, and a complete protease and phosphatase inhibitor cocktail. Lysates were then centrifuged at 4°C for 20 min at 13,500$g$, and the supernatants were collected. Subsequently, the supernatants were combined with anti-Flag resin for 2 h of binding. After binding, the resin was collected and washed three times with pull-down buffer (50 mM Tris, pH 8.0, 150 mM NaCl, 10% glycerol, 0.1% NP40), followed by boiling the resin with 1× SDS loading dye for sample preparation. The resulting protein sample were separated using SDS–PAGE, and the gel was cut for subsequent LC-MS/MS analysis.

### RNA sequencing

Total RNA was extracted from the following cell lines: PANC1, PANC1-shPRMT1, and PANC1-shPRMT1 rescued with PRMT1 oligomer, PRMT1 dimer, and PRMT1 monomer. The expression levels of PRMT1 in these cell lines were comparable with the endogenous PRMT1 levels in untreated PANC1 cells. RNA extraction was performed using Trizol according to the manufacturer's instructions. The construction of the RNA library and sequencing were carried out by Sangon Biotech in Shanghai. Differentially expressed genes (DEGs) were identified using the following cut-off criteria: |Log$_2$ (fold change)| ≥ 1.5 and $P$-value < 0.05.

### Quantification and statistical analysis

Statistical analysis was conducted using GraphPad Prism 8. For comparisons among more than two groups, one-way ANOVA followed by Tukey's multiple comparison tests was used. In cases involving two independent variables, two-way ANOVA with Tukey's multiple comparison test was used. All statistical analyses were performed using two-tailed tests. Plotted values are expressed as the mean ± SD. $P$-values below 0.05 were considered statistically significant (*$P$ < 0.05, **$P$ < 0.01, ***$P$ < 0.001). Pathway enrichment was assessed using GO and Kyoto Encyclopedia of Genes and Genomes (KEGG) tools in Hiplot Pro (https://hiplot.com.cn/) and Reactome gene sets in Metascape (https://metascape.org/).

# Data Availability

RNA sequencing data have been deposited in the Gene Expression Omnibus (GEO). The accession numbers (PRJNA1230539) are provided in the Deposited Data section of the Key Resources Table in Supplemental Data 1. In addition, all mass spectrometry raw files have been submitted to ProteomeXchange via the PRIDE database. Accessions numbers (PXD061743) for these files are also listed in the Deposited Data section of the Key Resources Table in Supplemental Data 1. The cryo-EM data collection statistics, including

parameters for raw micrographs, final density map, and refined atomic coordinates, have been deposited in the EMDB (accession code: EMD-61691) and the PDB (ID: 9JP0).

## Supplementary Information

## Acknowledgements

This work was supported by the National Natural Science Foundation of China (32270638, 32100464), the National Key Research and Development Program of China (2023YFE0118000), the Shenzhen Science and Technology Innovation Commission (JCYJ20230807091204009), the Fujian Provincial Natural Science Foundation of China (2022J01051), the Fundamental Research Funds for the Central Universities (20720220122), and the Nanqiang Outstanding Young Talents Program from Xiamen University. The Cryo-EM Facility of Southern University of Science and Technology (SUSTech) for technical assistance on cryo-EM data collection and Exit Postdoctoral Fellow for Shenzhen Research Funding (K22227502).

### Author Contributions

Y Ru: methodology and formal analysis.
X Zhou: methodology and visualization.
X Wang: methodology and formal analysis.
W Sun: methodology and visualization.
Y He: methodology.
G Hu: methodology.
W Li: methodology.
D Hu: methodology.
M Jiang: methodology.
Z Su: methodology and visualization.
F Niu: methodology and visualization.
G Chen: resources and methodology.
J Zeng: supervision and visualization.
S-F Sui: resources, supervision, and investigation.
W Liu: resources, supervision, and investigation.
Y Li: resources, supervision, validation, investigation, and writing—original draft and project administration.
S Chen: conceptualization, resources, supervision, funding acquisition, validation, investigation, methodology, project administration, and writing—original draft, review, and editing.

### Conflict of Interest Statement

The authors declare that they have no conflict of interest.

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
