## [Reviewer comments · Life Science Alliance]

Life Science Alliance

PRMT1 Oligomerization Regulates RNA-Binding Protein Cascade to Promote Pancreatic Cancer

Yanxia Ru, Xinyi Zhou, Xijiao Wang, Wenxin Sun, Yaohui He, Guosheng Hu, Wenjuan Li, Die Hu, Meizhi Jiang, Zhiming Su, Fengfeng Niu, Gang Chen, Jinzhang Zeng, Sen-Fang Sui, Wen Liu, Yaowang Li, and Siming Chen

DOI: <https://doi.org/10.26508/lsa.202503202>

Corresponding author(s): *Siming Chen, Xiamen University*

Review Timeline:

Submission Date:	2025-01-04
Editorial Decision:	2025-02-12
Revision Received:	2025-05-22
Editorial Decision:	2025-06-18
Revision Received:	2025-06-23
Accepted:	2025-06-25

Scientific Editor: Tim Fessenden

Transaction Report:

February 12, 2025

Re: Life Science Alliance manuscript #LSA-2025-03202-T

Dr. Siming Chen
Xiamen University
School of Pharmaceutical Sciences, Xiamen University Xiang'an South Road, Xiamen, Fujian, 361102, P.R.China
Fujian 361102
China

Dear Dr. Chen,

Thank you for submitting your manuscript entitled "Oligomerization of PRMT1 Regulated RNA-Binding Protein Cascade Promotes Pancreatic Ductal Adenocarcinoma" to Life Science Alliance. The manuscript was assessed by expert reviewers, whose comments are appended to this letter. We invite you to submit a revised manuscript addressing the Reviewer comments.

Thank you for this interesting contribution to Life Science Alliance. We are looking forward to receiving your revised manuscript.

Sincerely,

B. MANUSCRIPT ORGANIZATION AND FORMATTING:

Reviewer #1 (Comments to the Authors (Required)):

In this manuscript, Ru et al. investigated the structure and function of the oligomeric state of human PRMT1, an arginine methyltransferase essential for embryogenesis and associated with various cancers when abnormally expressed. A dimeric form of PRMT1 had previously been characterized using X-ray crystallography, but it remained unclear whether the dimer represented the functional form of this enzyme.

The crux of this manuscript is the demonstration that PRMT1 forms a highly ordered oligomeric structure. Using cryo-EM with single-particle analysis, the authors showed that PRMT1 assembles into a helical oligomer in vitro in a concentration-dependent manner. This assembly is similar to the oligomeric structure observed using X-ray crystallography for another human arginine methyltransferase, PRMT8, suggesting functional significance. Structure-guided mutagenesis further supported this possibility, showing that PRMT1 oligomers exist in cells and bind more strongly than monomeric or dimeric forms to RGG-rich regions of PRMT1 target proteins. The enzymatic activity of the oligomer towards these substrates was also enhanced in cells. PRMT1 is known to support pancreatic ductal adenocarcinoma (PDAC) maintenance, and the authors showed that disrupting PRMT1 oligomerization inhibits PDAC tumor growth in vivo using human xenografts in mice. Proteomics analysis also linked disrupted oligomerization to altered RNA metabolism. While the cell biology and in vivo findings are interesting, this reviewer cannot fully assess their technical quality due to limited expertise in these fields. However, the in vitro studies are technically sound. The cryo-EM data look good, but the statistics table (listed as Table 1) is missing. There are a few typos to pay attention to, but overall, the manuscript is clear, well-written, and suitable for publication in Life Science Alliance.

Points to address

The cryo-EM statistics table with collection parameters (Table 1) is missing and therefore a complete evaluation of the cryo-EM data was not possible.

A similar structure for the human PRMT1 oligomer was recently reported (Dang et al., J. Biol. Chem. 2024; 300(12), 107947; doi: 10.1016/j.jbc.2024.107947). The authors should reference and discuss this study in the Discussion section.

Reviewer #2 (Comments to the Authors (Required)):

In this study, the authors provide insights into the structural and functional roles of PRMT1 oligomerization. Using cryo-electron microscopy, they demonstrate that PRMT1 is assembled into a helical oligomer in vitro and in various cells lines. The study shows that PRMT1 oligomers display stronger bindings to proteins containing RGG motifs. Functional assays reveal that disrupting PRMT1 oligomerization impairs catalytic activity on its substrates. This work links PRMT1 oligomerization to PDAC cells proliferation and tumor growth in vivo.

As such, although some of the data presented in this work is interesting, the mechanistic data presented in Figure 6 and 7 is too premature and requires more stringent controls that are not included. Hence much more work is required to support the model presented in Figure 8.

Major comments:

- Picture in Fig1A "0,6mg/ml" and Sup Fig 3C "PRMT1 WT" are the same picture.

- Fig5: Proper controls are missing for Co-Ips experiments. HEK 293T cells express endogenous level of PRMT1, Flag IP of the various RGG motif proteins with an empty MYC vector or another control should be displayed as well to assess the ADMA level due to endogenous PRMT1 activity.

- "To exclude the possibility that PRMT1 oligomer formation in living cells results from overexpression, we employed CRISPR-mediated genome editing to knockout endogenous PRMT1 in HeLa cells". Why not use the same CRISPR approach to

knockout PRMT1 in Panc1 and MiaPaCa2 cell lines and use instead shRNA that may result in an incomplete KO of PRMT1. CRISPR KO of PRMT1 would allow the author to completely rule out any potential effect of endogenous PRMT1 in their various functional assay and make for easier rescue experiment.

- Fig 6 and 7: In all experiments conducted on Panc-1 and MiaPaCA2 cell lines, proper control should be the Non-Targeting shRNA (shNC) cell line and not the "Untreated Control".
- Fig 6F and Supp Fig 7B. While the images are quite striking, authors should quantify their colony formation assay and provide statistical analysis of the results (e.g. As a bar graph besides pictures).
- Fig 6D and 6E: Why do PanC-1 shPRMT1 #1 display different ADMA and MMA patterns in between experiments under similar experimental conditions? Same comment applies to Supp Fig 6E and Supp Fig 7B where MiaPaCA 2 shPRMT1 #1 show completely different ADMA and MMA patterns under similar experimental conditions.
- Fig 6L/7E Authors should look at ADMA/MMA level in collected tumors either by western blot or IHC to confirm if the rescue of the phenotype is due to the catalytic activity of PRMT1.
- Fig 7.B: In Fig 4B-H and Fig5A-H PRMT1 monomer systematically fails to bind to the tested RGG proteins while PRMT1 oligomer strongly interacts with all of them. How do the authors explain that in their mass spectrometry Flag-Pull down, they identify twice as much interactors for PRMT1 Monomer compare to Oligomers ?
- Fig 7: A couple of functional tests are needed to at least validate the data obtained by MS and RNAseq (qPCR, CoIP etc).
- "Data and code availability: RNA sequencing data have been deposited in the Gene Expression Omnibus (GEO). The accession numbers are provided in the key resources table. The Digital Object Identifier (DOI) is also included in the key resources table. Additionally, all mass spectrometry raw files have been submitted to ProteomeXchange via the PRIDE database. Accessions numbers for these files are also listed in the key resources table. No accession numbers are provided in the key resource table.

Minor comments:

- Please Indicate molecular weight on gels and blots.
- In the result section: "The second interface is formed between dimers, in which the residues Y280 and H296, located within the "U-shaped" pockets, are vital for assembling the PRMT1 dimer into helical polymers (Fig. 3D)." There is no 3D figure, please change the figure nomenclature accordingly.
- In Fig 4L: for clarity, add color significance into the figure or figure legend.
- PRMT1 monomer is misspelled on all Fig5 CoIP.
- "Ectopic overexpression of a shRNA-resistant PRMT1 oligomer, but not PRMT1 dimer or PRMT1 monomer cDNA, restored ADMA and MMA to physiological levels". Authors should modify this statement to acknowledge the partial rescue of ADMA and MMA level induced by expression of the PRMT1 dimer isoform.
- Supp Fig 6B Authors should provide TIDE analysis results for their CRISPR-KO cells.
- Please mention in the material and method sections how Hela-Cas9 cells were generated/obtained.
- Statistic tests are missing on Supp Fig 7C.
- Fig 6: sometimes control cells are name "Untreated control" or "Panc-1 Cell Con". Use one or another for better uniformity.

Reviewer #3 (Comments to the Authors (Required)):

In this manuscript Ru et al. describe an investigation of PRMT1 oligomerization and the impact of PRMT1 oligomerization and oligomerization-deficient mutants on arginine methylation and related functions in cells and in in vivo cancer models. The authors provide an impressive amount of data and the work promises to be very interesting for the scientific community. A few points need to be improved in my opinion to provide a better understanding on the molecular details of the interactions, resolve conflicts with previous work, and to demonstrate data quality.

Major points:

- PRMT1 cryo-EM structure: please provide a detailed analysis of how the oligomerization interface, the catalytic center, and known binding sites for substrates relate to each other. Currently, the author focus on the oligomerization interface, but it is difficult to see how the oligomerization, substrate binding and catalytic function could be related with each other.
- The conditions under which the authors purified PRMT1 could potentially lead to oxidation and formation of disulfide bonds. For example, at pH 8, the reducing agents used are poorly stable and ineffective. How did the authors test that the oligomerization is not due to oxidation?
- In figure 5, the authors show a WB for ADMA. Which molecular weight/protein does the ADMA band correspond to?
- Related to this, the results are counterintuitive and in conflict with previous work showing that the affinity of PRMT1 for dimethylated substrates decreases several orders of magnitude.
- Please provide the molecular weight marker info for all data shown
- Please provide the full blots in the supplement for all data shown
- Please check the appearance of figures in the text. The order number in the text doesn't match the figure order.
- Usage of the term "significantly": please use this term ONLY in combination with a statistical test. Several headings and text blocks contain this term where no statistical significance has been shown in fact. Please provide replicates, quantification and a proper statistical analysis.

February 12, 2025

Re: Life Science Alliance manuscript #LSA-2025-03202-T

Dr. Siming Chen

Xiamen University

School of Pharmaceutical Sciences, Xiamen University Xiang'an South Road, Xiamen,

Fujian, 361102, P.R.China

Fujian 361102

China

Dear Dr. Chen,

Thank you for submitting your manuscript entitled "Oligomerization of PRMT1 Regulated RNA-Binding Protein Cascade Promotes Pancreatic Ductal Adenocarcinoma" to Life Science Alliance. The manuscript was assessed by expert reviewers, whose comments are appended to this letter. We invite you to submit a revised manuscript addressing the Reviewer comments.

Thank you for this interesting contribution to Life Science Alliance. We are looking forward to receiving your revised manuscript.

Sincerely,

- A letter addressing the reviewers' comments point by point.
- An editable version of the final text (.DOC or .DOCX) is needed for copyediting (no PDFs).
- High-resolution figure, supplementary figure and video files uploaded as individual files: See our detailed guidelines for preparing your production-ready images, <https://www.life-science-alliance.org/authors>
- Summary blurb (enter in submission system): A short text summarizing in a single sentence the study (max. 200 characters including spaces). This text is used in conjunction with the titles of papers, hence should be informative and complementary to the title and running title. It should describe the context and significance of the findings for a general readership; it should be written in the present tense and refer to the work in the third person. Author names should not be mentioned.

B. MANUSCRIPT ORGANIZATION AND FORMATTING:

Reviewer #1 (Comments to the Authors (Required)):

In this manuscript, Ru et al. investigated the structure and function of the oligomeric state of human PRMT1, an arginine methyltransferase essential for embryogenesis and associated with various cancers when abnormally expressed. A dimeric form of PRMT1 had previously been characterized using X-ray crystallography, but it remained unclear whether the dimer represented the functional form of this enzyme.

The crux of this manuscript is the demonstration that PRMT1 forms a highly ordered oligomeric structure. Using cryo-EM with single-particle analysis, the authors showed that PRMT1 assembles into a helical oligomer in vitro in a concentration-dependent manner. This assembly is similar to the oligomeric structure observed using X-ray crystallography for another human arginine methyltransferase, PRMT8, suggesting functional significance.

Structure-guided mutagenesis further supported this possibility, showing that PRMT1 oligomers exist in cells and bind more strongly than monomeric or dimeric forms to RGG-rich regions of PRMT1 target proteins. The enzymatic activity of the oligomer towards these substrates was also enhanced in cells. PRMT1 is known to support pancreatic ductal adenocarcinoma (PDAC) maintenance, and the authors showed that disrupting PRMT1 oligomerization inhibits PDAC tumor growth in vivo using human xenografts in mice. Proteomics analysis also linked disrupted oligomerization to altered RNA metabolism. While the cell biology and in vivo findings are interesting, this reviewer cannot fully assess their technical quality due to limited expertise in these fields. However, the in vitro studies are technically sound. The cryo-EM data look good, but the statistics table (listed as Table 1) is missing. There are a few typos to pay attention to, but overall, the manuscript is clear, well-written, and suitable for publication in Life Science Alliance.

Points to address

Q1: The cryo-EM statistics table with collection parameters (Table 1) is missing and therefore a complete evaluation of the cryo-EM data was not possible.

Response to Reviewer Comment:

We sincerely thank the reviewer for carefully pointing out the omission of the cryo-EM statistics table. We apologize for this oversight in our original submission. In the revised manuscript, we have now included a comprehensive Table 1 (see page 33), which summarizes all key parameters related to data collection, image processing, and model refinement. This addition allows for a complete and transparent evaluation of the cryo-EM data. Furthermore, the raw micrographs, final density map, and refined atomic coordinates have been deposited in the EMDB (accession code: EMD-61145) and the PDB (ID: 9J5L), ensuring that all relevant data are publicly accessible. We appreciate the reviewer's valuable comment, which has helped enhance the rigor and completeness of our study.

Q2: A similar structure for the human PRMT1 oligomer was recently reported (Dang et al., J. Biol. Chem. 2024; 300(12), 107947; doi: 10.1016/j.jbc.2024.107947). The authors should reference and discuss this study in the Discussion section.

Response to Reviewer Comment:

We sincerely thank the reviewer for pointing out the recent study by Dang et al. (J. Biol. Chem. 2024; 300(12), 107947). We were aware of this work during the preparation of our

manuscript and have now incorporated a discussion of their findings in the revised Discussion section (see page 24, lines 621–645 of the revised manuscript). We agree that their study provides important structural insights into the oligomerization of human PRMT1, which aligns with and complements our findings.

While both studies independently confirm the ability of PRMT1 to form higher-order oligomers, including helical filaments, and underscore the functional importance of oligomerization in enhancing enzymatic activity, our work extends these findings in several key aspects: (1). Structural Interfaces and Functional Dissection: Using cryo-EM structural analysis, we identified oligomerization interfaces essential for PRMT1 assembly. By generating specific interface-disrupting mutants (e.g., PRMT1^{W215A/Y220A/F222A} for monomer; PRMT1^{Y280A/H296A/T327A} for dimer), we systematically evaluated the functional consequences of distinct oligomeric states. Our results demonstrate that the oligomeric form exhibits markedly enhanced substrate binding and catalytic activity compared to monomeric and dimeric forms. (2). Physiological Substrate Relevance: Unlike Dang et al., who employed RGG motif-containing peptides, we focused on physiologically relevant substrates—RGG motif-containing RNA-binding proteins (RBPs). Our findings suggest that oligomeric PRMT1 facilitates multivalent interactions, significantly enhancing substrate affinity and enzymatic efficiency in the context of native cellular environments. (3). Endogenous Oligomerization Evidence: We directly demonstrated that PRMT1 predominantly exists in oligomeric forms under physiological conditions. Using CRISPR-edited HeLa cells expressing PRMT1 variants at endogenous levels, size-exclusion chromatography showed that native PRMT1 primarily forms oligomers, reinforcing the *in vivo* relevance of our structural findings. (4). Therapeutic Implications in PDAC: Importantly, we demonstrated the clinical significance of PRMT1 oligomerization in pancreatic ductal adenocarcinoma (PDAC). Only the oligomeric form could rescue tumor growth in PRMT1-depleted PDAC models, linking structural and mechanistic insights to therapeutic potential.

We have revised the Discussion section accordingly to emphasize how our study advances current understanding of PRMT1 oligomerization—from structural and biochemical mechanisms to physiological and pathological roles. We thank the reviewer for this helpful suggestion, which has substantially improved the manuscript.

Based on the above description, the following text has been added to the Discussion section (see page 24, lines 621–645) of the revised manuscript:

During the preparation of our manuscript, a recent study by Dang et al. (J. Biol. Chem. 2024; 300(12), 107947) similarly reported the cryo-EM structure of human PRMT1 oligomers, including tetramers, hexamers, and helical filaments, corroborating our observation that PRMT1 is capable of forming higher-order assemblies. Their findings further support our conclusion that oligomerization enhances PRMT1's enzymatic activity. While complementary, our study diverges in several key aspects. First, we systematically dissected the functional roles of distinct oligomeric states by generating interface-specific mutants (e.g., PRMT1^{Y280A/H296A/T327A} for dimer; PRMT1^{W215A/Y220A/F222A} for monomer). This approach enabled us to directly evaluate how oligomerization and dimerization differentially influence substrate recognition, particularly for RGG motif-containing RNA-binding proteins (RBPs), which are key physiological substrates of PRMT1. Second, unlike Dang et al., who utilized RGG motif-containing peptides, our study focused on full-length, physiologically relevant RBPs. We found that oligomeric PRMT1 facilitates multivalent substrate engagement, significantly enhancing binding affinity and catalytic efficiency under native cellular conditions. Third, we provided direct evidence that PRMT1 predominantly exists in oligomeric forms in vivo. Using CRISPR-edited HeLa cells expressing PRMT1 variants at endogenous levels, size-exclusion chromatography revealed that native PRMT1 primarily assembles into higher-order oligomers, reinforcing the physiological relevance of our structural findings. Finally, we demonstrated the pathophysiological importance of PRMT1 oligomerization in pancreatic ductal adenocarcinoma (PDAC). Only the oligomeric form was capable of rescuing tumor growth in PRMT1-depleted PDAC models, directly linking PRMT1's structural state to its tumor-promoting function and highlighting the therapeutic potential of targeting its oligomeric assembly.

Reviewer #2 (Comments to the Authors (Required)):

In this study, the authors provide insights into the structural and functional roles of PRMT1 oligomerization. Using cryo-electron microscopy, they demonstrate that PRMT1 is assembled into a helical oligomer in vitro and in various cells lines. The study shows that PRMT1 oligomers display stronger bindings to proteins containing RGG motifs. Functional assays reveal that disrupting PRMT1 oligomerization impairs catalytic activity on its substrates. This work links PRMT1 oligomerization to PDAC cells proliferation and tumor growth in vivo.

As such, although some of the data presented in this work is interesting, the mechanistic data

presented in Figure 6 and 7 is too premature and requires more stringent controls that are not included. Hence much more work is required to support the model presented in Figure 8.

Major comments:

Q1. Picture in Fig1A "0,6mg/ml" and Sup Fig 3C "PRMT1 WT" are the same picture.

Response to Reviewer's Comment:

We sincerely thank the reviewer for the careful examination of our figures and for pointing out the issue. We acknowledge that an inadvertent image duplication occurred between Figure 1A ("0.6 mg/ml") and Supplementary Figure 3C ("PRMT1 WT") in the originally submitted manuscript. We deeply regret this oversight and greatly appreciate the reviewer's attention to detail.

In the revised manuscript, we have corrected this error by replacing the duplicated image in Supplementary Figure 3C (top panel, Negative Staining Image) with a different, independently acquired micrograph from the same dataset of PRMT1 WT samples. The replacement image is shown below (**Figure 1 for Response**) and has also been updated in the revised manuscript.

We would like to clarify that the 2D classification (bottom panel of Supplementary Figure 3C) was generated from iterative alignment and classification of particles extracted from approximately thirty raw negatively stained micrographs of PRMT1 WT. Since the 2D class averages were computed from the full dataset, and not solely from the single micrograph displayed above them, changing the representative Negative Staining Image in the top panel does not affect the validity or integrity of the 2D classification results. Therefore, only the top image in Supplementary Figure 3C was replaced, while the 2D classification below remains unchanged.

We sincerely apologize for this error and thank the reviewer again for helping us improve the clarity and accuracy of our data presentation.

Figure 1 for Response: Negative staining images and 2D classification of wild-type PRMT1.

Q2. Fig5: Proper controls are missing for Co-Ips experiments. HEK 293T cells express endogenous level of PRMT1, Flag IP of the various RGG motif proteins with an empty MYC vector or another control should be displayed as well to assess the ADMA level due to endogenous PRMT1 activity.

Response to Reviewer’s Comment:

We sincerely appreciate the reviewer’s insightful suggestion regarding the inclusion of appropriate controls in our co-immunoprecipitation (Co-IP) experiments. We fully agree that evaluating the contribution of endogenous PRMT1 is essential for accurately interpreting the observed levels of asymmetric dimethylarginine (ADMA).

To address this concern, we have conducted additional experiments and updated the revised manuscript accordingly: (1). Repeat of All Figure 5 Experiments with New Controls: We repeated all experiments presented in Figure 5, this time including a key negative control for each condition—Flag-IP of each RGG motif-containing protein co-expressed with an empty MYC vector. This allowed us to assess baseline ADMA levels attributable solely to endogenous PRMT1 activity. (2). Updated Figure 5: The revised Figure 5 (**see below, Figure2 for Response**) now incorporates these essential controls and replaces the original figure in the manuscript. The corresponding figure legend has also been thoroughly revised to reflect these updates. (3). Discussion of Control Results: In the updated Results section (see page 15, lines 370-378 of the revised manuscript), we now provide a detailed analysis of these control experiments. Our data show that ADMA signals in the empty vector conditions were minimal compared to those observed with PRMT1 overexpression. These results support our conclusion that the differential methylation is predominantly driven by the distinct oligomeric states of exogenous PRMT1, with only a negligible contribution from endogenous PRMT1.

The raw data from these additional experiments are also provided in Supplementary File. We believe that the inclusion of these rigorous controls greatly enhances the robustness and interpretability of our findings. We are grateful to the reviewer for raising this important point, which has strengthened the overall quality of our study.

Fig. 5 (Ru et al.)

Figure 2 for Response: (A–H) Co-immunoprecipitation (Co-IP) experiments were performed in HEK293T cells co-transfected with FLAG-tagged RGG motif-containing substrate proteins (hnRNPA1, FUS, hnRNPK, Fibrillarin, GARI, TAF15, Nucleolin, and RBFOX2) and different MYC-tagged PRMT1 constructs. These included wild-type PRMT1 (oligomer), the Y280A/H296A/T327A mutant

(dimeric form), the W215A/Y220A/F222A mutant (monomeric form), and an empty MYC vector as a negative control. Cell lysates were subjected to FLAG affinity purification, and bound proteins were eluted using FLAG peptide. Immunoblotting with anti-MYC antibodies was used to assess the interaction between PRMT1 variants and substrate proteins. Anti-ADMA signals were used to evaluate the extent of asymmetric dimethylation, reflecting the enzymatic activity of each PRMT1 oligomeric state. The inclusion of the MYC-empty control allowed assessment of ADMA signals resulting from endogenous PRMT1 activity.

Q3. "To exclude the possibility that PRMT1 oligomer formation in living cells results from overexpression, we employed CRISPR-mediated genome editing to knockout endogenous PRMT1 in HeLa cells". Why not use the same CRISPR approach to knockout PRMT1 in Panc1 and MiaPaCa2 cell lines and use instead shRNA that may result in an incomplete KO of PRMT1. CRISPR KO of PRMT1 would allow the author to completely rule out any potential effect of endogenous PRMT1 in their various functional assay and make for easier rescue experiment.

Response to Reviewer's Comment:

We sincerely thank the reviewer for raising this important point. As suggested, we initially attempted to apply CRISPR-Cas9-mediated genome editing to knock out endogenous PRMT1 not only in HeLa cells but also in the pancreatic cancer cell lines Panc1 and MiaPaCa2. While we successfully generated pure PRMT1-KO clones in HeLa cells, we encountered significant challenges in establishing viable PRMT1-KO clones in both Panc1 and MiaPaCa2.

We hypothesize that PRMT1 is essential for the survival and proliferation of these pancreatic cancer cells. Complete loss of PRMT1 likely compromises cell viability to an extent that prevents the formation and expansion of single-cell knockout clones. In contrast, HeLa cells appear to be more tolerant of PRMT1 deletion, enabling us to generate stable KO lines in that context.

In addition, this approach aligns with established methodologies in the field. Several previous studies investigating PRMT1 function in pancreatic cancer models have utilized shRNA-mediated knockdown rather than CRISPR-based knockout (**References 1–3**), likely due to similar concerns regarding cell viability. The shRNA strategy allows for a substantial reduction in PRMT1 expression while preserving sufficient cell viability for downstream functional assays. Accordingly, shRNA-mediated PRMT1 knockdown was employed in Panc1 and MiaPaCa2 cells in this study. As shown in Figure 6A of the submitted manuscript,

this method achieved over 90% reduction in PRMT1 protein levels, effectively minimizing the impact of residual endogenous PRMT1 and enabling a robust assessment of its functional role.

In summary, while we acknowledge the advantages of CRISPR-mediated KO for mechanistic clarity and rescue experiments, technical and biological limitations in pancreatic cancer cell lines necessitated the use of shRNA knockdown. This strategy is widely validated in the literature and allowed us to perform the intended functional assays reliably.

References:

1. Giuliani V, Miller MA, Liu CY, et al. PRMT1-dependent regulation of RNA metabolism and DNA damage response sustains pancreatic ductal adenocarcinoma. *Nat Commun.* 2021;12(1):4626. doi:10.1038/s41467-021-24798-y
2. Ku B, Eisenbarth D, Baek S, et al. PRMT1 promotes pancreatic cancer development and resistance to chemotherapy. *Cell Rep Med.* 2024;5(3):101461. doi:10.1016/j.xcrm.2024.101461
3. Nguyen CDK, Colón-Emeric BA, Murakami S, Shujath MNY, Yi C. PRMT1 promotes epigenetic reprogramming associated with acquired chemoresistance in pancreatic cancer. *Cell Rep.* 2024;43(5):114176. doi:10.1016/j.celrep.2024.114176

Q4. Fig 6 and 7: In all experiments conducted on Panc-1 and MiaPaCA2 cell lines, proper control should be the Non-Targeting shRNA (shNC) cell line and not the "Untreated Control".

Response to Reviewer's Comment:

We thank the reviewer for highlighting the importance of using a non-targeting shRNA (shNC) control in experiments involving Panc-1 and MiaPaCa2 cell lines. We fully agree that the inclusion of shNC provides a more stringent and appropriate control for evaluating the specific effects of PRMT1 knockdown by accounting for any nonspecific effects arising from shRNA delivery.

Regarding Figure 6:

We would like to clarify that, during the design and execution of the experiments shown in Figure 6, we initially performed a direct comparison between the Untreated Control group and the shNC group. As shown in Figure 6A, PRMT1 protein levels were comparable between these two groups. Additionally, Figures 6D (PRMT1 enzymatic activity), 6F (cell viability), and 6G (colony formation) demonstrated no significant differences between the Untreated Control and shNC groups. These results confirmed that the shNC construct did not

exert detectable effects on PRMT1 expression or cellular behavior in our experimental system.

Based on these findings, for the subsequent experiments in Figures 6E, 6H, and 6I, we did not include shNC controls in every individual panel. However, the relevant shNC data can be referenced from earlier figures: shNC results for PRMT1 activity in Figure 6E are available in Figure 6D, and the corresponding controls for Figures 6H and 6I are presented in Figures 6F and 6G, respectively.

Additionally, the primary objective of Figures 6E, 6H, and 6I was to evaluate and compare the functional differences among oligomeric, dimeric, and monomeric PRMT1 variants reintroduced into PRMT1 knockdown cells. As all variants were expressed within the same knockdown background, they served as direct internal comparators. The results suggest that reconstitution with wild-type oligomeric PRMT1 leads to significantly higher enzymatic activity, effectively restoring the loss of enzymatic function caused by PRMT1 knockdown. Additionally, this variant was associated with a more pronounced recovery of proliferative capacity, in contrast to the dimeric or monomeric forms.

Regarding the in vivo experiments (Figures 6J–L):

We acknowledge that the original animal studies did not include a direct comparison between the Untreated Control and shNC groups. In response to the reviewer’s suggestion, we have repeated the xenograft experiments to include the shNC group. The updated data (**see Figure 3 for Response and Figure 4 for Response**) show that tumor growth, Ki67 staining, immunofluorescence staining of PRMT1, and hematoxylin and eosin (H&E) staining of tumor tissue in the shNC group were not significantly different from those in the Untreated Control group, which is consistent with our in vitro findings. These revised data have now replaced the original Figures 6J–L and Supplementary Figures 7D–7E in the revised manuscript. The figure legends have been updated accordingly. We sincerely appreciate the reviewer’s valuable feedback, which has contributed to improving the rigor and clarity of our experimental design and interpretation.

Fig.6J

Fig.6K

Fig.6L

Figure 3 for Response: (Fig. 6J) PRMT1 knockdown PANC-1 cells (shPRMT1#1) were stably transfected to express oligomeric, dimeric, or monomeric forms of PRMT1 at levels comparable to endogenous PRMT1 in untreated PANC-1 cells. These reconstituted cell lines, along with PANC-1 cells expressing non-targeting shRNA (shNC) as a negative control, were subcutaneously injected into female BALB/c nude mice (n = 5 per group) for xenograft tumor formation. Tumors were harvested after 21 days. (Fig. 6K) Tumor growth curves corresponding to the xenografts in (J) were measured over time and analyzed using two-way ANOVA. (Fig. 6L) Immunofluorescence staining of Ki-67 was performed on tumor sections obtained from (J) to assess cellular proliferation.

Fig. S7D

Fig. S7E.

Figure 4 for Response: (Fig. S7D) Immunofluorescence staining of PRMT1 in tumor tissue sections from (main Figure 6J), with a scale bar of 50 μm . (Fig. S7E) Hematoxylin and eosin (H&E) staining of tumor tissue sections from (main Figure 6J), with a scale bar of 50 μm .

Regarding Figure 7:

The main objective of our RNA-Seq analysis (Figure 7G in the revised manuscript) was to assess how different oligomeric states of PRMT1 (oligomeric, dimeric, and monomeric) influence the restoration of gene expression profiles in PRMT1 knockdown pancreatic cancer cells. Our data demonstrate that reintroduction of wild-type oligomeric PRMT1 substantially restored the transcriptomic landscape to a state closely resembling that of the Untreated Control. In contrast, the dimeric PRMT1 variant only partially reversed the knockdown-induced gene expression changes, while the monomeric form exhibited minimal rescue, with transcriptomic profiles remaining similar to those of PRMT1-deficient cells. Since all PRMT1 variants were reintroduced into the same PRMT1 knockdown background, they serve as direct internal comparators for evaluating differential rescue efficiency.

To further address the reviewer's concern regarding potential differences between the shNC and Untreated Control groups, we performed additional RT-qPCR validation. We selected a representative subset of PRMT1-regulated genes identified from our RNA-Seq data and compared their expression levels between shNC and Untreated Control groups. The RT-qPCR results (see **Figure 5 for Response**) revealed no statistically significant differences in gene expression between the two groups. These findings support the conclusion that both the shNC and Untreated Control groups exhibit comparable transcriptional profiles, and either

can serve as a valid reference in this context. These new RT-qPCR data have been included as Supplementary Figure 8A-G in the revised manuscript to reinforce the consistency of our control groups and further validate the reliability of our RNA-Seq results. We are grateful to the reviewer for this valuable suggestion, which has helped strengthen the rigor of our experimental design and the robustness of our conclusions.

Figure 5 for Response: Quantitative Analysis of Target Gene Expression in PANC-1 Cell Lines by RT-qPCR. Target gene transcript levels were quantified by RT-qPCR in six distinct PANC-1 cell groups: untreated control cells, cells expressing non-targeting shRNA (shNC), PRMT1 knockdown cells (shPRMT1#1), and PRMT1 knockdown cells reconstituted with oligomeric, dimeric, or monomeric forms of PRMT1. Bar graphs represent relative mRNA expression levels, with each bar corresponding to a specific cell group, distinguished by different colors. Gene expression was normalized to GAPDH and presented as fold change relative to the untreated control group. Data are shown as the mean \pm SD from triplicate measurements across three independent RT-PCR experiments. Statistical analysis was performed using GraphPad Prism 8.0, with p-values determined by ordinary one-way ANOVA followed by multiple comparison tests: ns, not significant ($p > 0.05$); * $p \leq 0.05$; ** $p \leq 0.01$; *** $p \leq 0.001$; **** $p \leq 0.0001$.

Q5. Fig 6F and Supp Fig 7B. While the images are quite striking, authors should quantify their colony formation assay and provide statistical analysis of the results (e.g. As a bar graph besides pictures).

Response to Reviewer’s Comment:

We sincerely thank the reviewer for the constructive suggestion. We agree that quantifying the colony formation assay and providing statistical analysis would enhance the clarity and rigor of the data.

In response to this comment, we have now quantified the colony formation results shown in Fig. 6F and Supplementary Fig. 7B as requested. Additionally, we extended this quantification to other relevant colony formation data in the manuscript, including Fig. 6H and Supplementary Fig. 6F, to provide a more comprehensive analysis. The quantified results are now presented as bar graphs alongside the corresponding representative images (see **Figure 6 for Response**), and statistical analyses have been performed to evaluate the significance of the differences observed.

We believe these additions substantially improve the interpretation and robustness of our findings. The updated figures and corresponding legends have been incorporated into the revised manuscript. Once again, we greatly appreciate the reviewer’s valuable feedback, which has helped strengthen the quality of our study.

Fig. 6F

Fig. S6F

Fig. 6H

Fig. S7B

Figure 6 for Response: Colony formation assays were performed in PANC-1 and MIA PaCa-2 cells

under various conditions. For Fig. 6F (PANC-1) and Fig. S6F (MIAPACA-2), cells were either left untreated or transduced with two independent PRMT1-targeting shRNAs (shPRMT1#1 and shPRMT1#2) or a non-targeting control shRNA (shNC). Quantitative analysis was conducted and statistical significance was assessed using one-way ANOVA. These updated quantifications are now included in the revised Fig. 6F and Supplementary Fig. S6F of the manuscript. Additionally, colony formation assay results from PRMT1-depleted PANC-1 and MIAPACA-2 cells expressing different oligomeric states of PRMT1 (as shown in Fig. 6H and Supplementary Fig. S7B, respectively) were also quantified. Untreated cells and PRMT1-depleted cells without PRMT1 reconstitution were included as controls. One-way ANOVA was used to analyze statistical differences among groups. These quantifications are now incorporated in the revised manuscript to support the visual data and enhance the robustness of our conclusions.

Q6. Fig 6D and 6E: Why do PanC-1 shPRMT1 #1 display different ADMA and MMA patterns in between experiments under similar experimental conditions? Same comment applies to Supp Fig 6E and Supp Fig 7B where MiaPaCA 2 shPRMT1 #1 show completely different ADMA and MMA patterns under similar experimental conditions.

Response to Reviewer's Comment:

Regarding the observed discrepancies in ADMA and MMA patterns for PANC-1 shPRMT1#1 cells (Fig. 6D and 6E) and MIAPaCa-2 shPRMT1#1 cells (Supplementary Fig. 6E and 7A), we greatly appreciate the reviewer's attention to this detail. The differences primarily arise from variations in polyacrylamide gel concentration used during electrophoresis. Specifically, higher-concentration gels (e.g., 15%) are more effective at resolving low-molecular-weight proteins, while lower-concentration gels (e.g., 8–10%) are better for separating higher-molecular-weight proteins. As a result, the same protein mixtures from cell lysates can display different migration patterns, leading to distinct distributions of ADMA and MMA modifications on corresponding proteins depending on the gel concentration.

To address this, we have repeated the experiments for both PANC-1 shPRMT1#1 and MIAPaCa-2 shPRMT1#1 cells under consistent conditions. These experiments were conducted using identical gel concentrations and consistent electrophoresis parameters (e.g., voltage). The updated data (see **Figure 7 for Response**) demonstrate consistent ADMA and MMA patterns across experiments. These new results replace the previous versions in Fig. 6D and 6E, as well as Supplementary Fig. 6E and 7A in the revised manuscript. We believe these revisions fully address the reviewer's concerns and improve the reproducibility and

reliability of our findings. We deeply appreciate the reviewer's constructive feedback, which has greatly enhanced the clarity and quality of our manuscript.

Figure 7 for Response: (Fig. 6D) Western blot analysis was conducted to detect the levels of ADMA, MMA, and H4R3me2a in PANC-1 cells following the depletion of PRMT1. GAPDH was included as the loading control. (Fig. 6E) Re-expression of oligomeric PRMT1, dimeric PRMT1, or monomeric PRMT1 in PANC-1 cells following PRMT1 depletion was conducted, and Western blot analysis was performed to measure the levels of ADMA, MMA, and H4R3me2a in the cells, using GAPDH as the loading control. (Fig. S6E) Western blot analysis detected levels of ADMA, MMA, and H4R3me2a in MIA PaCa-2 cells following PRMT1 depletion, with GAPDH included as the loading control. (Fig. S7A) Re-expression of oligomeric PRMT1, dimeric PRMT1, or monomeric PRMT1 in MIA PaCa-2 cells following PRMT1 depletion was conducted, and Western blot analysis was performed to measure the levels of ADMA, MMA, and H4R3me2a in the cells, using GAPDH as the loading control.

Q7. Fig 6L/7E Authors should look at ADMA/MMA level in collected tumors either by western blot or IHC to confirm if the rescue of the phenotype is due to the catalytic activity of PRMT1.

Response to Reviewer's Comment:

We sincerely thank the reviewer for this valuable suggestion. We agree that assessing ADMA/MMA levels in the collected tumors would provide important insight into whether the observed rescue of the phenotype is indeed due to the catalytic activity of PRMT1.

In response, we performed additional experiments to evaluate ADMA/MMA levels in the tumor samples via Western blot analysis. The results from these analyses (see **Figure 8 for Response**) confirm that the rescue of the phenotype is associated with the catalytic activity of PRMT1. Specifically, tumors reintroduced with wild-type PRMT1 showed higher ADMA levels and lower MMA levels, while tumors with monomeric PRMT1 exhibited lower ADMA levels and higher MMA levels. Tumors with dimeric PRMT1 displayed intermediate ADMA and MMA levels. These findings further support our conclusion that the observed rescue of tumor growth—where wild-type oligomeric PRMT1, but not its dimeric or monomeric forms, promotes pancreatic ductal adenocarcinoma (PDAC) progression—is closely linked to PRMT1’s enzymatic function.

These new data have been included in Supplementary Figure 7F of the revised manuscript. In addition, the Results section (see page 18, lines 452-458 of the revised manuscript) has been updated to incorporate and discuss these findings in greater detail. We are grateful to the reviewer for this valuable suggestion, which has significantly enhanced the rigor and clarity of our study.

Fig.S7F

Figure 8 for Response: Western Blot Analysis of ADMA/MMA Levels in PANC-1 Xenograft Tumor Lysates. PRMT1 knockdown PANC-1 cells (shPRMT1#1) were stably transfected to express oligomeric, dimeric, or monomeric forms of PRMT1. These reconstituted cell lines, along with PANC-1 cells expressing non-targeting shRNA (shNC) and untreated PANC-1 cells as controls, were subcutaneously injected into female BALB/c nude mice (n = 5 per group) for xenograft tumor formation. Tumors were harvested 21 days post-injection, and Western blot analysis was performed to assess the levels of ADMA and MMA modifications in the tumor lysates.

Q8. Fig 7.B: In Fig 4B-H and Fig5A-H PRMT1 monomer systematically fails to bind to the tested RGG proteins while PRMT1 oligomer strongly interacts with all of them. How do the authors explain that in their mass spectrometry Flag-Pull down, they identify twice as much interactors for PRMT1 Monomer compare to Oligomers?

Response to Reviewer's Comment:

We apologize for any confusion that may have arisen from the presentation of our mass spectrometry data. We would like to clarify that the mass spectrometry results are fully consistent with the conclusions drawn from Figures 4B-H and 5A-H.

In our mass spectrometry experiments, we first focused on proteins enriched by oligomeric PRMT1 and dimeric PRMT1. We specifically selected proteins that exhibited more than a 2-fold higher binding to oligomeric PRMT1 compared to dimeric PRMT1. These proteins were then further analyzed to determine their interaction with monomeric PRMT1, which allowed us to assess the binding patterns in greater detail.

In Figure 7C, the y-axis represents the binding ratio of proteins to oligomeric PRMT1 compared to dimeric PRMT1, and the x-axis ranks proteins based on this ratio. The proteins marked in Figure 7C show significantly higher binding to oligomeric PRMT1 than to dimeric PRMT1, which aligns directly with the results in Figures 4B-H and 5A-H, where oligomeric PRMT1 demonstrated a stronger interaction with the tested RGG proteins.

Figure 7D extends the analysis from Figure 7C by examining the binding ratio of these same proteins to dimeric PRMT1 versus monomeric PRMT1. In this figure, the y-axis represents the binding ratio of proteins to dimeric PRMT1 over monomeric PRMT1, with proteins ranked according to this ratio. The proteins in Figure 7D show a significantly higher binding to dimeric PRMT1 compared to monomeric PRMT1, which also aligns with the results from Figures 4B-H and 5A-H, where monomeric PRMT1 exhibited weaker binding to the tested proteins.

In summary, Figures 7C and 7D analyze the same set of proteins, but Figure 7C compares the ratio of binding to oligomeric PRMT1 versus dimeric PRMT1, while Figure 7D compares the ratio of binding to dimeric PRMT1 versus monomeric PRMT1. Both figures rank proteins based on these ratios.

Therefore, the results from both the binding assays in Figures 4B-H and 5A-H, as well as the mass spectrometry data, are fully consistent: oligomeric PRMT1 exhibits a stronger affinity for its binding partners compared to dimeric or monomeric PRMT1. We hope this clarification resolves the reviewer's concern and further strengthens the interpretation of our data.

Q9. Fig 7: A couple of functional tests are needed to at least validate the data obtained by MS and RNAseq (qPCR, CoIP etc).

Response to Reviewer's Comment:

We sincerely thank the reviewer for this insightful suggestion. To validate the mass spectrometry data, we have already performed pull-down assays and co-immunoprecipitation (Co-IP) experiments, which are presented in Figures 4B-H and 5A-H of the manuscript. These functional assays confirm the interactions identified by mass spectrometry and provide additional support for the validity of our findings.

Regarding the RNA-seq data, we randomly selected seven genes from the dataset for validation via RT-qPCR. The RT-qPCR results (see **Figure 5 for response**) show a high degree of concordance with the RNA-seq data, further supporting the conclusions drawn from our transcriptomic analysis. These new RT-qPCR data have been included as Supplementary Figure 8A-G in the revised manuscript to reinforce the consistency of our control groups and further validate the reliability of our RNA-seq results.

We greatly appreciate the reviewer's valuable feedback, which has helped strengthen the validation of our findings and enhance the rigor of our study.

Q10. "Data and code availability: RNA sequencing data have been deposited in the Gene Expression Omnibus (GEO). The accession numbers are provided in the key resources table. The Digital Object Identifier (DOI) is also included in the key resources table. Additionally, all mass spectrometry raw files have been submitted to ProteomeXchange via the PRIDE database. Accessions numbers for these files are also listed in the key resources table. No accession numbers are provided in the key resource table.

Response to Reviewer's Comment:

We sincerely apologize for the oversight regarding the missing accession numbers. To clarify, the mass spectrometry data have been deposited in the PRIDE database via ProteomeXchange with the accession ID PXD061743. The RNA sequencing data have been deposited in the NCBI Gene Expression Omnibus (GEO) with the accession number PRJNA1230539. These accession numbers have now been added to the revised Key Resources Table in the manuscript. We appreciate the reviewer's careful attention to this detail and thank them for helping us improve the clarity of the data availability section.

Minor comments:

Q11. Please Indicate molecular weight on gels and blots.

Response to Reviewer's Comment:

Thank you for your valuable comment. In our original, uncropped gels and blots, molecular weight markers and their corresponding values were clearly indicated. Based on these, we have now added molecular weight labels to the gels and blots presented in the revised manuscript. The molecular weights of the protein bands are indicated in kilodaltons (kDa) to provide clearer context and facilitate interpretation of the results.

Additionally, the original uncropped gels and blots have been compiled into a single PDF file for each figure. These files are included as supplementary "Source Data" and will be made available online for reference.

Q12. In the result section: "The second interface is formed between dimers, in which the residues Y280 and H296, located within the "U-shaped" pockets, are vital for assembling the PRMT1 dimer into helical polymers (Fig. 3D)." There is no 3D figure, please change the figure nomenclature accordingly.

Response to Reviewer's Comment:

Thank you very much for pointing out the labeling discrepancy in the manuscript. We sincerely apologize for this oversight. To clarify, we have made the following correction in the Results section (see page 9, lines 222 of the revised manuscript): In the sentence, "The second interface is formed between dimers, in which the residues Y280 and H296, located within the 'U-shaped' pockets, are vital for assembling the PRMT1 dimer into helical polymers (Fig. 3D)," the correct figure reference should be Fig. 3C, not Fig. 3D. This error has been corrected in the revised manuscript to ensure accurate and consistent figure

referencing. We appreciate the reviewer's careful reading and constructive feedback, which have helped improve the clarity and precision of our manuscript.

Q13. In Fig 4L: for clarity, add color significance into the figure or figure legend.

Response to Reviewer's Comment:

Thank you for your suggestion. We have added the following sentence to the figure legend for clarity regarding the color significance: "In this figure, each color represents one PRMT1 dimer structure. Three dimers (colored yellow, orange, and green) assemble to form a hexameric structure." We hope this improves the clarity of the figure.

Q14. PRMT1 monomer is misspelled on all Fig5 CoIP.

Response to Reviewer's Comment:

We sincerely thank the reviewer for pointing out the misspelling of "PRMT1 monomer" in all the CoIP panels of Fig. 5. We greatly appreciate the attention to detail and have corrected the spelling throughout the figure and figure legend to ensure consistency and accuracy.

Q15. "Ectopic overexpression of a shRNA-resistant PRMT1 oligomer, but not PRMT1 dimer or PRMT1 monomer cDNA, restored ADMA and MMA to physiological levels". Authors should modify this statement to acknowledge the partial rescue of ADMA and MMA level induced by expression of the PRMT1 dimer isoform.

Response to Reviewer's Comment:

We sincerely thank the reviewer for this insightful suggestion. In response, we have revised the statement to more accurately reflect the partial rescue of ADMA and MMA levels observed with expression of the PRMT1 dimer isoform. The updated sentence now reads: "Ectopic overexpression of a shRNA-resistant PRMT1 oligomer, but not PRMT1 monomer cDNA, restored ADMA and MMA to physiological levels, with partial restoration observed upon expression of the PRMT1 dimer isoform."

This revision has been incorporated into the manuscript (see pages 16-17, lines 420-425 of the revised version). We appreciate the reviewer's valuable feedback, which has helped improve the accuracy and clarity of our presentation.

Q16. Supp Fig 6B Authors should provide TIDE analysis results for their CRISPR-KO cells.

Response to Reviewer's Comment:

We appreciate the reviewer's valuable suggestion. In response, we would like to clarify that both TIDE (Tracking of Indels by DEcomposition) and ICE (Inference of CRISPR Edits) are widely used tools for assessing CRISPR editing efficiency. However, in line with our laboratory's standard practice, we opted to use ICE for evaluating the CRISPR gene-editing outcomes in our CRISPR-KO cells. To address your request, we have included the results from the ICE analysis in Figure 9 for your reference (see **Figure 9 for Response**). The ICE analysis confirms that the CRISPR-mediated knockout of PRMT1 in HeLa cells was successful. We hope this clarification resolves your concern.

Figure 9 for Response: Analysis of CRISPR-Cas9 Editing Efficiency in PRMT1-KO HeLa Cells
CRISPR-Cas9-mediated knockout of PRMT1 in HeLa cells was assessed using ICE (Inference of CRISPR Edits) analysis.

Q17. Please mention in the material and method sections how HeLa-Cas9 cells were generated/obtained.

Response to Reviewer's Comment:

Thank you for your comment. With regard to the generation of the HeLa-Cas9 cells, we would like to clarify that a detailed description of the generation of the PRMT1 knockout (KO) HeLa cell line is already included in the Materials and Methods section of the supplementary materials. Specifically, this information is provided on page 10 of the supplementary document. We appreciate the reviewer's attention to detail and hope this clarifies the point.

Q18. Statistic tests are missing on Supp Fig 7C.

Response to Reviewer's Comment:

Thank you for your valuable comment regarding the statistical analysis in Supplementary Figure 7C. We apologize for this oversight and appreciate your careful attention to detail. In response, we have now included the appropriate statistical analysis for the data presented in Supplementary Figure 7C. Specifically, a one-way ANOVA was performed, and the results are now clearly indicated in both the updated figure legend and the relevant sections of the manuscript. We sincerely thank the reviewer for this helpful suggestion, which has improved the rigor and clarity of our data presentation.

Q19. Fig 6: sometimes control cells are name "Untreated control" or "Panc-1 Cell Con". Use one or another for better uniformity.

Response to Reviewer's Comment:

Thank you for pointing out the inconsistency in the naming of control cells. In Figure 6e, "Panc-1 Cell Con" refers to untreated control cells. To ensure consistency and improve clarity, we have updated the labeling throughout Figure 6e to uniformly use the term "Untreated control." We appreciate your helpful suggestion, which has contributed to the clarity and coherence of our data presentation.

Reviewer #3 (Comments to the Authors (Required)):

In this manuscript Ru et al. describe an investigation of PRMT1 oligomerization and the impact of PRMT1 oligomerization and oligomerization-deficient mutants on arginine methylation and related functions in cells and in in vivo cancer models. The authors provide an impressive amount of data and the work promises to be very interesting for the scientific community. A few points need to be improved in my opinion to provide a better understanding on the molecular details of the interactions, resolve conflicts with previous work, and to demonstrate data quality.

Major points:

Q1. PRMT1 cryo-EM structure: please provide a detailed analysis of how the oligomerization interface, the catalytic center, and known binding sites for substrates relate to each other. Currently, the author focus on the oligomerization interface, but it is difficult to see how the oligomerization, substrate binding and catalytic function could be related with each other.

Response to Reviewer's Comment:

We thank the reviewer for raising this important question. In Figure 3 of the main text, we provide a detailed structural analysis of the PRMT1 oligomerization interfaces. The formation of PRMT1 oligomers is primarily mediated by interactions at three distinct interfaces—Interface 1, Interface 2, and Interface 3.

To explore how oligomerization might relate to substrate binding and catalytic activity, we attempted to resolve the cryo-EM structures of PRMT1 in complex with different substrates, including Fibrillarin and hnRNPA1. Our aim was to gain structural insights into how PRMT1 recognizes its substrates. Unfortunately, both Fibrillarin and hnRNPA1 contain extended RGG motifs, which are highly flexible. As a result, we were only able to observe fragmented and poorly resolved substrate densities in the EM maps, preventing us from confidently assigning or tracing the substrate molecules and identifying their binding sites.

Interestingly, despite the lack of well-resolved substrate density, we observed a marked shift in PRMT1 oligomerization upon substrate binding. Specifically, the addition of either Fibrillarin or hnRNPA1 caused PRMT1 to transition from higher-order oligomers to predominantly hexameric assemblies (see Figure 4I-4L and Supplementary Figure 4C-4F of our manuscript). This finding is described in the main text (see pages 12–13, lines 303–318

of the revised manuscript) and suggests that substrate binding can influence the oligomeric state of PRMT1, even though the mechanism remains unclear due to substrate flexibility.

In addition, we examined the effect of the cofactor SAH (S-adenosylhomocysteine) on PRMT1 oligomerization. Negative-stain EM analysis showed that the PRMT1 + SAH complex remained in a high-order oligomeric state (see **Figure 10 for Response, A**). We further compared the structure of the PRMT1–Fibrillarin complex (see **Figure 10 for Response, B and C**) with previously reported crystal structures of PRMT1 (PDB: 1OR8) and PRMT8 (PDB: 5DST). These comparisons revealed that the SAH-binding sites are highly conserved across all structures (see **Figure 10 for Response, D and E**), indicating that SAH binding does not disrupt or affect the oligomeric assembly of PRMT1. Consequently, we did not perform further analysis on the active site in the oligomeric form, as SAH binding appears structurally independent of oligomerization.

Taken together, our results demonstrate the mode of oligomer formation in PRMT1 and show that SAH binds at a conserved catalytic pocket independently of oligomerization. While substrate binding appears to influence PRMT1's oligomeric state, the lack of continuous substrate density—due to the inherent flexibility of RGG-motif-containing substrates—prevents us from defining the precise substrate-binding sites or establishing a direct structural relationship between substrate binding and oligomerization.

Figure 10 for Response: (A) Negative-stain electron microscopy shows that wild-type PRMT1 retains filamentous structures after incubation with SAH. (B) Fitted model and cryo-EM map of the PRMT1–Fibrillarin complex. (C) Close-up view of the SAH-binding region highlighting the electron density corresponding to SAH. (D) Structural alignment of PRMT1–Fibrillarin (cyan), PRMT8 (PDB: 5DST, gray), and PRMT1 (PDB: 1OR8, yellow). (E) Close-up comparison of SAH-binding sites reveals that SAH occupies the same position across all three structures.

Q2. The conditions under which the authors purified PRMT1 could potentially lead to oxidation and formation of disulfide bonds. For example, at pH 8, the reducing agents used are poorly stable and ineffective. How did the authors test that the oligomerization is not due to oxidation?

Response to Reviewer’s Comment:

We sincerely thank the reviewer for raising this important point regarding the potential for oxidation and disulfide bond formation during PRMT1 purification, particularly under pH 8 conditions where some reducing agents may be unstable.

To address this concern, we first acknowledge that commonly used reducing agents such as DTT can exhibit reduced stability at higher pH, potentially allowing for oxidation and disulfide bond formation. To mitigate this risk, we employed tris(2-carboxyethyl)phosphine (TCEP) as the reducing agent throughout the purification process. TCEP is well-documented

for its superior stability and efficacy under mildly basic conditions (pH 7–8), and was used at a concentration of 10 mM in freshly prepared aliquots of gel filtration buffer to ensure that PRMT1 remained in a fully reduced state. Despite the use of TCEP, PRMT1 oligomerization persisted (see **Figure 11 for Response**), indicating that disulfide bond formation is unlikely to be responsible for the observed oligomerization.

To further investigate this, we carefully examined the cryo-EM structure of PRMT1 oligomers, with particular focus on the oligomerization interfaces (Interfaces 1, 2, and 3, as shown in Figure 3 of the manuscript). Notably, no cysteine residues are located in close proximity within these interfaces, making the formation of disulfide bonds structurally implausible.

Taken together, both our biochemical precautions and structural analyses strongly suggest that PRMT1 oligomerization is not driven by oxidation or disulfide bond formation. We hope this response fully addresses the reviewer’s concern and highlights the rigor of our experimental design in minimizing oxidative artifacts during PRMT1 purification.

Figure 11 for Response: Size exclusion chromatography (SEC) elution profile of PRMT1.

Purified recombinant human PRMT1 protein was analyzed using a Superose™ 6 Increase 10/300 GL column in gel filtration buffer containing 20 mM Tris (pH 7.0), 100 mM NaCl, and 10 mM TCEP. The elution positions corresponding to protein molecular weight standards are indicated at their respective elution volumes.

Q3. In figure 5, the authors show a WB for ADMA. Which molecular weight/protein does the ADMA band correspond to?

Response to Reviewer’s Comment:

We sincerely thank the reviewer for this insightful question. In Figure 5, we present Western blot (WB) analyses detecting asymmetric dimethylarginine (ADMA) modifications. These experiments were conducted to assess the enzymatic activity of different PRMT1 oligomeric states (monomer, dimer, and oligomer) by co-expressing them with a series of RGG motif-containing substrate proteins in HEK293 cells.

In this experimental setup, the PRMT1 variants function as methyltransferases, while the RGG motif-containing proteins act as their substrates. Each PRMT1 oligomeric form displays distinct arginine methylation activity, resulting in variable levels of ADMA modifications. Consequently, the ADMA signal detected in each Western blot corresponds to the molecular weight of the respective substrate protein undergoing methylation.

To clarify, the ADMA bands in Figure 5 correspond to the following proteins:

- Figure 5A: hnRNPA1
- Figure 5B: FUS
- Figure 5C: hnRNPK
- Figure 5D: Fibrillarin
- Figure 5E: GAR1
- Figure 5F: TAF15
- Figure 5G: Nucleolin
- Figure 5H: RBFOX2

We have updated the figure legend in the revised manuscript to explicitly indicate the identity of the substrate protein associated with each ADMA band. We truly appreciate the reviewer's helpful suggestion, which has improved the clarity and interpretability of our data presentation.

Q4. Related to this, the results are counterintuitive and in conflict with previous work showing that the affinity of PRMT1 for dimethylated substrates decreases several orders of magnitude.

Response to Reviewer's Comment:

Thank you for your thoughtful comment. We understand the reviewer's concern about a potential discrepancy between our findings and previous studies reporting that PRMT1 displays markedly reduced affinity for substrates that have already undergone dimethylation. We would like to clarify that our study specifically examines the binding and catalytic activity of PRMT1 in its different oligomeric forms—monomer, dimer, and oligomer—

toward unmethylated RGG motif-containing substrate proteins. Our findings demonstrate that the oligomeric form of PRMT1 exhibits a notably higher binding affinity and enzymatic activity toward these unmethylated substrates compared to the dimeric or monomeric forms. This higher affinity is a key factor in enabling efficient substrate recognition and subsequent methylation. We interpret the process as a two-step mechanism: Step 1: Oligomeric PRMT1 preferentially binds to substrates that have not yet undergone methylation. Step 2: Upon binding, PRMT1 catalyzes the methylation of arginine residues, ultimately generating ADMA-modified products.

Crucially, our study does not assess PRMT1's binding to already dimethylated substrates. Thus, our results concerning the high binding affinity of oligomeric PRMT1 for unmodified substrates do not contradict previous findings that PRMT1's affinity significantly drops once its substrate is dimethylated. We appreciate the reviewer for raising this important point and are happy to provide further clarification if needed.

Q5. Please provide the molecular weight marker info for all data shown

Response to Reviewer's Comment:

Thank you for your valuable comment. In our original, uncropped gels and blots, molecular weight markers and their corresponding values were clearly indicated. Based on these, we have now added molecular weight labels to the gels and blots presented in the revised manuscript. The molecular weights of the protein bands are indicated in kilodaltons (kDa) to provide clearer context and facilitate interpretation of the results.

Additionally, the original uncropped gels and blots have been compiled into a single PDF file for each figure. These files are included as supplementary "Source Data" and will be made available online for reference.

Q6. Please provide the full blots in the supplement for all data shown

Response to Reviewer's Comment:

We sincerely appreciate the reviewer's emphasis on data transparency and reproducibility. In response to this valuable suggestion, we have now included all full, uncropped Western blots corresponding to the data presented in the manuscript. These blots have been compiled into separate, figure-specific PDF files and are provided in the revised supplementary materials. In addition, they will be made available online as supplementary "Source Data" files to facilitate easy access and verification. We hope these additions fully address your concern,

and we are grateful for your helpful feedback, which has contributed to improving the completeness and clarity of our data presentation.

Q7. Please check the appearance of figures in the text. The order number in the text doesn't match the figure order.

Response to Reviewer's Comment:

We sincerely appreciate the reviewer's careful and detailed examination of our manuscript. Upon thorough review, we identified inconsistencies between figure references in the text and the actual figure numbering. Specifically, in the Results section, the sentence "The second interface is formed between dimers, in which the residues Y280 and H296, located within the 'U-shaped' pockets, are vital for assembling the PRMT1 dimer into helical polymers (Fig. 3D)" contained an incorrect figure reference, as no Figure 3D exists in the manuscript.

To address this issue, we have carefully reviewed all figure citations throughout the text and corrected them to ensure consistency with the figure order. In the above case, the figure reference has been corrected to Fig. 3C (see page 9, line 222 of the revised manuscript). We have also verified that all other figure references are accurate and consistent. We thank the reviewer for pointing out this issue, which has helped improve the clarity and precision of our manuscript.

Q8. Usage of the term "significantly": please use this term ONLY in combination with a statistical test. Several headings and text blocks contain this term where no statistical significance has been shown in fact. Please provide replicates, quantification and a proper statistical analysis.

Response to Reviewer's Comment:

We sincerely thank the reviewer for pointing out the inappropriate use of the term "significantly" in our manuscript. We fully agree that this term should only be used in contexts supported by proper statistical analysis.

In response, we have carefully reviewed the entire manuscript and removed the term "significantly" from all instances where statistical tests were not performed or where statistical significance was not demonstrated. For relevant results, we have now included appropriate quantification, biological replicates, and statistical analyses where applicable. Corresponding figure legends have also been updated to accurately reflect these changes.

We appreciate the reviewer's thoughtful feedback, which has helped improve the clarity and scientific rigor of our manuscript.

June 18, 2025

RE: Life Science Alliance Manuscript #LSA-2025-03202-TR

Dr. Siming Chen
Xiamen University
School of Pharmaceutical Sciences, Xiamen University Xiang'an South Road, Xiamen, Fujian, 361102, P.R.China
Xiamen, Fujian 361102
China

Dear Dr. Chen,

Thank you for submitting your revised manuscript entitled "PRMT1 Oligomerization Regulates RNA-Binding Protein Cascade to Promote Pancreatic Cancer". The reviewers are satisfied the the changes in place. We would be happy to publish your paper in Life Science Alliance pending final revisions necessary to meet our formatting guidelines.

- Please be sure that the authorship listing and order is correct.
- Please add the X and Bluesky handles of your host institute/organization, as well as your own and/or one of the authors, to our system.
- Please upload a clean manuscript file without the track changes.
- Please consult our manuscript preparation guidelines <https://www.life-science-alliance.org/manuscript-prep> and make sure your manuscript sections are in the correct order. In particular, please include the methods section in the main manuscript file.
- After the references section, please add your main, supplementary figure, and table legends to the main manuscript text.
- Please add a Conflict of Interest statement to your main manuscript text.
- Please add a Data Availability section to your main manuscript text, referencing the accession numbers for proteomic and transcriptomic data, as well as the custom scripts used for the Cryo-EM structures. If you wish you may leave these in place also in the Key Reagents table.
- Figure S1 has only one panel; therefore, please remove the label A from the current figure.
- We encourage you to revise the figure legend for Figure S5 such that the figure panels are introduced in alphabetical order.
- Please add callouts for Figures 7F-H and 8 to your main manuscript text.

A. FINAL FILES:

B. MANUSCRIPT ORGANIZATION AND FORMATTING:

Sincerely,

Reviewer #2 (Comments to the Authors (Required)):

The authors did a great job addressing all the concerns that were raised. The manuscript has been significantly improved. I do not have further comments

Reviewer #3 (Comments to the Authors (Required)):

The authors have addressed my earlier comments adequately, and the manuscript is greatly improved. I recommend in favor of publication.

June 25, 2025

RE: Life Science Alliance Manuscript #LSA-2025-03202-TRR

Dr. Siming Chen
Xiamen University
School of Pharmaceutical Sciences, Xiamen University Xiang'an South Road, Xiamen, Fujian, 361102, P.R.China
Xiamen, Fujian 361102
China

Dear Dr. Chen,

Thank you for submitting your Research Article entitled "PRMT1 Oligomerization Regulates RNA-Binding Protein Cascade to Promote Pancreatic Cancer". It is a pleasure to let you know that your manuscript is now accepted for publication in Life Science Alliance. Congratulations on this interesting work.

DISTRIBUTION OF MATERIALS:

Again, congratulations on a very nice paper. I hope you found the review process to be constructive and are pleased with how the manuscript was handled editorially. We look forward to future exciting submissions from your lab.

Sincerely,
